# HDAC1 SUMOylation promotes Argonaute-directed transcriptional silencing in *C. elegans*

**Heesun Kim[1†], Yue-He Ding[1†], Gangming Zhang[1], Yong-Hong Yan[2], Darryl Conte Jr[1], Meng-Qiu Dong[2], Craig C Mello[1,3]***

[1]RNA Therapeutics Institute, University of Massachusetts Medical School, Worcester, United States; [2]National Institute of Biological Sciences, Beijing, China; [3]Howard Hughes Medical Institute, Chevy Chase, United States

**Abstract** Eukaryotic cells use guided search to coordinately control dispersed genetic elements. Argonaute proteins and their small RNA cofactors engage nascent RNAs and chromatin-associated proteins to direct transcriptional silencing. The small ubiquitin-like modifier (SUMO) has been shown to promote the formation and maintenance of silent chromatin (called heterochromatin) in yeast, plants, and animals. Here, we show that Argonaute-directed transcriptional silencing in *Caenorhabditis elegans* requires SUMOylation of the type 1 histone deacetylase HDA-1. Our findings suggest how SUMOylation promotes the association of HDAC1 with chromatin remodeling factors and with a nuclear Argonaute to initiate de novo heterochromatin silencing.

***For correspondence:**
Craig.Mello@umassmed.edu

[†]These authors contributed equally to this work

**Competing interests:** The authors declare that no competing interests exist.

## Introduction

Argonautes are an ancient family of proteins that utilize short nucleic acid guides (usually composed of 20–30 nts of RNA) to find and regulate cognate RNAs (reviewed in *Meister, 2013*). Argonaute-dependent small RNA pathways are linked to chromatin-mediated gene regulation in diverse eukaryotes, including plants, protozoans, fungi, and animals (reviewed in *Martienssen and Moazed, 2015*). Connections between Argonautes and chromatin are best understood from studies in fission yeast *Schizosaccharomyces pombe*, where the Argonaute, Ago1, a novel protein, Tas3, and a heterochromatin protein 1 (HP1) homolog, Chp1, comprise an RNA-induced transcriptional silencing (RITS) complex that maintains and expands heterochromatin (*Verdel et al., 2004*). The Chp1 protein binds H3K9me3 through its conserved chromodomain (*Partridge et al., 2000*; *Partridge et al., 2002*) and is thought to anchor Ago1 within heterochromatin, where it is poised to engage nascent RNA transcripts (*Holoch and Moazed, 2015*). Low-level transcription of heterochromatin is thought to create a platform for propagating small-RNA amplification and heterochromatin maintenance (reviewed in *Holoch and Moazed, 2015*).

In yeast, a protein complex termed SHREC (Snf2/Hdac-containing Repressor complex) has been linked to both the establishment and maintenance of heterochromatin and transcriptional silencing (*Job et al., 2016*; *Motamedi et al., 2008*; *Sugiyama et al., 2007*). SHREC contains a homolog of type 1 histone deacetylase (HDAC), a homolog of Mi-2 and CHD3 ATP-dependent chromatin remodelers, and a Krüppel-type C2H2 zinc finger protein. SHREC therefore resembles the nucleosome remodeling and deacetylase (NuRD) complex in animals (*Denslow and Wade, 2007*; *Torchy et al., 2015*). NuRD complexes play a key role in converting chromatin from an active to a silent state, and can be recruited to targets through sequence-independent interactions (e.g., modified chromatin or methylated DNA) or through sequence-specific interactions via the Krüppel-type C2H2 zinc finger protein and other DNA-binding factors (*Ecco et al., 2017*; *Lupo et al., 2013*). In

animals, NuRD complexes function broadly in developmental gene regulation and transposon silencing (*Ecco et al., 2017*; *Feschotte and Gilbert, 2012*; *Ho and Crabtree, 2010*).

The post-translational modification of heterochromatin factors by the small ubiquitin-like protein SUMO has been implicated at several steps in the establishment and maintenance of transcriptional gene silencing and has been linked to silencing mediated by the Piwi Argonaute (reviewed in *Ninova et al., 2019*). The addition of SUMO (i.e., SUMOylation) requires a highly conserved E2 SUMO-conjugating enzyme, UBC9, which interacts with substrate-specific co-factor (E3) enzymes to covalently attach SUMO to lysines in target proteins (*Johnson, 2004*). Whereas ubiquitylation is primarily associated with the protein turnover (reviewed in *Glickman and Ciechanover, 2002*), SUMOylation is primarily associated with changes in protein interactions, especially with proteins that contain SUMO-interacting motifs (SIMs) (reviewed in *Kerscher, 2007*). In mammals, for example, SUMOylation of the KAP1 corepressor is required to recruit the NuRD complex and the SETDB histone methyltransferase via SIM domains in CHD3 and SETDB and to silence KRAB targets (*Ivanov et al., 2007*).

Here, we identify a connection between the SUMO pathway and transcriptional silencing initiated by the Piwi Argonaute pathway in the *Caenorhabditis elegans* germline. We show that SUMOylation of C-terminal lysines on the type 1 HDAC, HDA-1, is required for Piwi-mediated transcriptional silencing. SUMOylation of HDA-1 promotes its association with conserved components of the *C. elegans* NuRD complex, the nuclear Argonaute HRDE-1/WAGO-9, the histone demethylase SPR-5, and the SetDB-related histone methyltransferase MET-2. Our findings suggest how SUMOylation of HDAC1 promotes the recruitment and assembly of an Argonaute-guided chromatin remodeling complex that orchestrates de novo transcriptional gene silencing in the *C. elegans* germline.

## Results

### The SUMO and HDAC pathways promote piRNA silencing

In *C. elegans*, silencing initiated by the Piwi Argonaute PRG-1 depends on chromatin modifications at the target locus and on a group of worm-specific Argonautes (WAGOs), including nuclear-localized family members WAGO-9/HRDE-1 and WAGO-10 (*Ashe et al., 2012*; *Bagijn et al., 2012*; *Lee et al., 2012*; *Shirayama et al., 2012*) and nuage-localized family members WAGO-1 and WAGO-4 (*Gu et al., 2009*; *Shirayama et al., 2012*; *Xu et al., 2018*). WAGOs engage antisense guides produced by cellular RNA-dependent RNA polymerases (RdRPs) (*Gu et al., 2009*). How the downstream machinery that amplifies and maintains silencing is recruited to targets remains unknown. To identify additional components of the transcriptional silencing arm of the piRNA pathway, we performed an RNAi-based genetic screen of chromatin factors and modifiers using a sensor transgene silenced by the piRNA pathway (*Figure 1A*; see *Seth et al., 2018*). The piRNA sensor is 100% silenced in wild-type germlines, but is desilenced in the germlines of *prg-1(tm872)*, *rde-3 (ne3370)*, and *hrde-1/wago-9(ne4769)* mutants, resulting in expression of a bright, easily scored GFP::CSR-1 fusion protein (*Figure 1B*; *Seth et al., 2018*). Even the partial inactivation of known piRNA silencing factors activated sensor expression in a percentage of exposed animals (*Figure 1C* and *Supplementary file 1*).

Our RNAi screen identified many components of known HDAC complexes, as well as SUMO pathway factors (*Figure 1C* and *Supplementary file 1*). For example, depletion of *mep-1* (Krüppel-type zinc finger protein) and other genes encoding NuRD-complex co-factors (*let-418*/Mi-2, *hda-1*/HDAC1, *lin-40*/MTA2/3, *lin-53*/RBBP4/7, and *dcp-66*/GATAD2B) desilenced the piRNA sensor (*Figure 1C* and *Supplementary file 1*). Depletion of SIN3-HDAC complex genes, *sin-3* (SIN3) and *mrg-1* (MORF4L1), also desilenced the reporter (*Figure 1C* and *Supplementary file 1*). RNAi of two SUMO pathway genes, *smo-1* (SUMO) and *ubc-9* (SUMO-conjugating enzyme), desilenced the sensor. Notably, however, RNAi of the conserved E3 SUMO ligase gene *gei-17* (PIAS1/Su(var)2–10) (*Hari et al., 2001*; *Mohr and Boswell, 1999*; *Ninova et al., 2020*) did not desilence the piRNA sensor (*Figure 1C*).

Null alleles of many of these genes cause embryonic arrest, which precludes an analysis of silencing in the adult germline. To further explore the role of SUMO and HDAC factors in piRNA silencing, we therefore tested whether partial or conditional loss-of-function alleles activate the piRNA sensor. Auxin-inducible degron alleles of *hda-1*, *let-418*, *mep-1*, and *mrg-1* and a truncation allele of *sin-3* all

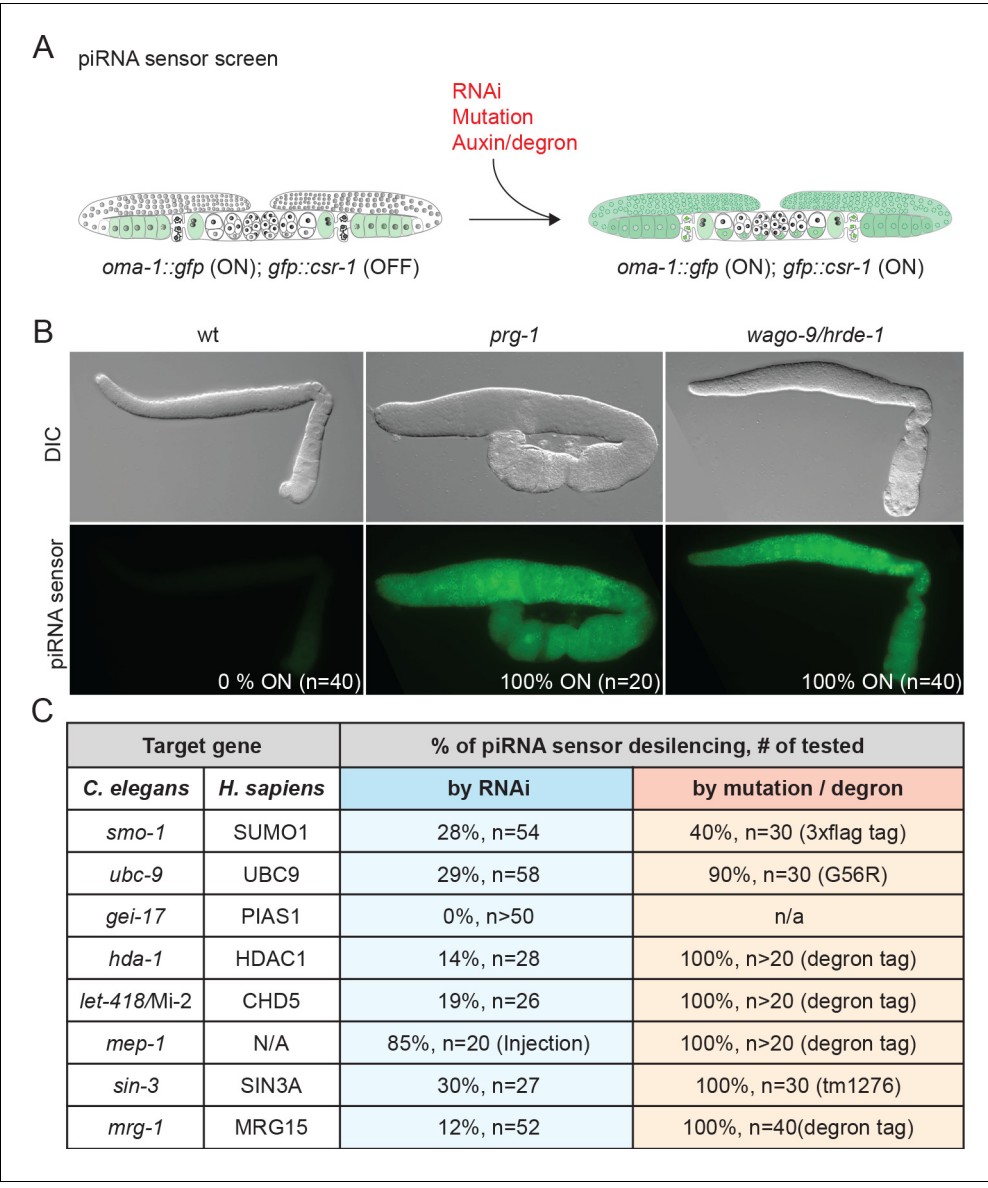

**Figure 1.** SUMOylation and chromatin remodeling factors promote piRNA-mediated silencing. (**A**) Schematic of the piRNA sensor screen. The piRNA sensor strain contains a *gfp::csr-1* transgene that is silenced by the piRNA pathway in the presence of an active *oma-1::gfp* transgene (**Seth et al., 2018**). OMA-1::GFP localizes to the cytoplasm of oocytes. Inactivation of the piRNA pathway (by RNAi, mutation, or auxin-inducible protein depletion) desilences the transgene, resulting in GFP::CSR-1 expression in perinuclear P-granules throughout the germline, as shown in (**B**). (**B**) Differential interference contrast and epifluorescence images of dissected gonads in wild-type (wt), *prg-1(tm872)*, and *wago-9/hrde-1(ne4769)* worms. PRG-1 is required to initiate silencing, while WAGO-9 is required to maintain silencing. The percentage of desilenced worms and number of worms scored are shown. (**C**) Analysis of SUMO and chromatin remodeling factors required for piRNA-mediated silencing. Genes identified in the RNAi-based screen of chromatin factors are listed with their human homologs and with the percentage of worms that express GFP::CSR-1 among the total number of worms analyzed (n) when function is reduced by RNAi (blue column) or by either mutation or degron-dependent protein depletion (peach column).

desilenced the piRNA sensor in 100% of worms examined (*Figure 1C*). We found that a *3xflag* fusion to the endogenous *smo-1* gene desilenced the piRNA sensor in ~40% of adults analyzed (*Figure 1C*), suggesting that this tagged *smo-1* allele behaves like a partial loss of function. A strain expressing a temperature-sensitive UBC-9(G56R) protein desilenced the sensor in 90% of the animals at the semi-permissive temperature of 23°C (*Figure 1C*) (*Kim et al., 2021*). By contrast,

presumptive null alleles of *gei-17* completely failed to desilence the piRNA sensor strain (*Kim et al., 2021*). Together, these findings suggest that histone deacetylase complexes and components of the SUMO pathway promote piRNA-mediated silencing.

## SUMOylation of HDA-1 promotes piRNA surveillance

In a parallel study, we found that *C. elegans* HDA-1 is SUMOylated in the adult germline (*Kim et al., 2021*). Moreover, we showed that SUMOylation of HDA-1, formation of an adult NuRD complex, and piRNA-mediated silencing depend redundantly on PIE-1, a CCCH zinc finger protein with SUMO E3-like function, and on GEI-17, a homolog of PIAS1/Su(var)2–10 SUMO E3 ligase, suggesting that SUMOylation of HDA-1 might promote piRNA surveillance (*Kim et al., 2021*). Human HDAC1 is SUMOylated near its C-terminus on consensus SUMO-acceptor lysines 444 and 476 (*Figure 2A*; *David et al., 2002*), but HDA-1 does not contain consensus SUMO acceptor sites, and poor conservation between the C-termini of human HDAC1 and worm HDA-1 did not suggest the identity of potential SUMO-acceptor sites. However, GPS-SUMO prediction software identified lysines 444 and 459 of HDA-1 as possible non-consensus SUMO-acceptor sites (*Figure 2A*; *Zhao et al., 2014*). To investigate whether one or both of these lysine residues is required for HDA-1 SUMOylation, we used CRISPR genome editing to mutate the endogenous *hda-1* gene in a strain that expresses SUMO fused to 10 N-terminal histidines, *his10::smo-1* (*Kim et al., 2021*). We then used nickel-nitrilotriacetic acid (Ni-NTA) affinity chromatography under stringent denaturing conditions (*Tatham et al., 2009*) to capture SUMOylated proteins from worm lysates. The Ni-column eluates were then analyzed by western blotting for HDA-1. SUMOylated HDA-1 was recovered by Ni-NTA affinity chromatography from wild-type lysate and from lysates of each single-site lysine-to-arginine mutant (*Figure 2B*, lanes 7–9), but was absent when both K444 and K459 were mutated together, HDA-1(KKRR) (*Figure 2B*, lane 10). As a control, the protein MRG-1, which is highly SUMOylated in wild-type worms (*Kim et al., 2021*), was readily detected in worms expressing HDA-1(KKRR) (*Figure 2B*). When we introduced the piRNA sensor into the HDA-1 SUMO-acceptor site mutants, we found that whereas the piRNA sensor remained silent in the single-site mutants (n = 30) (data not shown), it was expressed in 100% of HDA-1(KKRR) worms (*Figure 2C*).

Since the SUMOylation sites in HDA-1 are very close to the C-terminus, we wondered if appending SUMO via translation fusion to the HDA-1(KKRR) mutant protein might restore HDA-1 function in the piRNA sensor assay. Using CRISPR, we inserted a modified *smo-1* open reading frame just before the stop codon of the *hda-1[KKRR]* gene at the endogenous *hda-1* locus. The modified SMO-1 fusion cannot be transferred to other proteins because it lacks the GG amino acids required for conjugation (*Dorval and Fraser, 2006*; see Materials and methods). Surprisingly, the resulting *hda-1 [KKRR]::smo-1* strain was homozygous viable, healthy, and expressed an HDA-1::SMO-1 fusion protein at levels similar to those observed for wild-type HDA-1 (*Figure 2D*). Strikingly, appending SMO-1 to the C-terminus of HDA-1(KKRR) completely rescued the silencing defect of HDA-1(KKRR) (*Figure 2C*). Moreover, HDA-1(KKRR)::SMO-1 rescued the piRNA-mediated silencing defects of *smo-1* and *ubc-9* mutants (*Figure 2C*), suggesting that the SUMO pathway promotes piRNA-mediated silencing via C-terminal SUMOylation of HDA-1.

## HDA-1 SUMOylation is not required for maintenance of piRNA-initiated silencing

Once silencing is established by the upstream components of the Piwi pathway, it can be maintained indefinitely by the co-transcriptional arm of the pathway without the continued need for *prg-1* activity (*Shirayama et al., 2012*). For example, the initial silencing of the *gfp::cdk-1* transgene requires PRG-1 activity, but maintenance of silencing does not (*Shirayama et al., 2012*). To ask if HDA-1 SUMOylation is required for the maintenance of piRNA-initiated silencing, we crossed an already silenced *gfp::cdk-1* transgene into an *hda-1[KKRR]* mutant strain. We found that the *gfp-1::cdk-1* reporter remained silent even after five generations in HDA-1(KKRR) worms (n = 26), supporting the idea that C-terminal SUMOylation of HDA-1 is not required to maintain piRNA-induced transcriptional silencing.

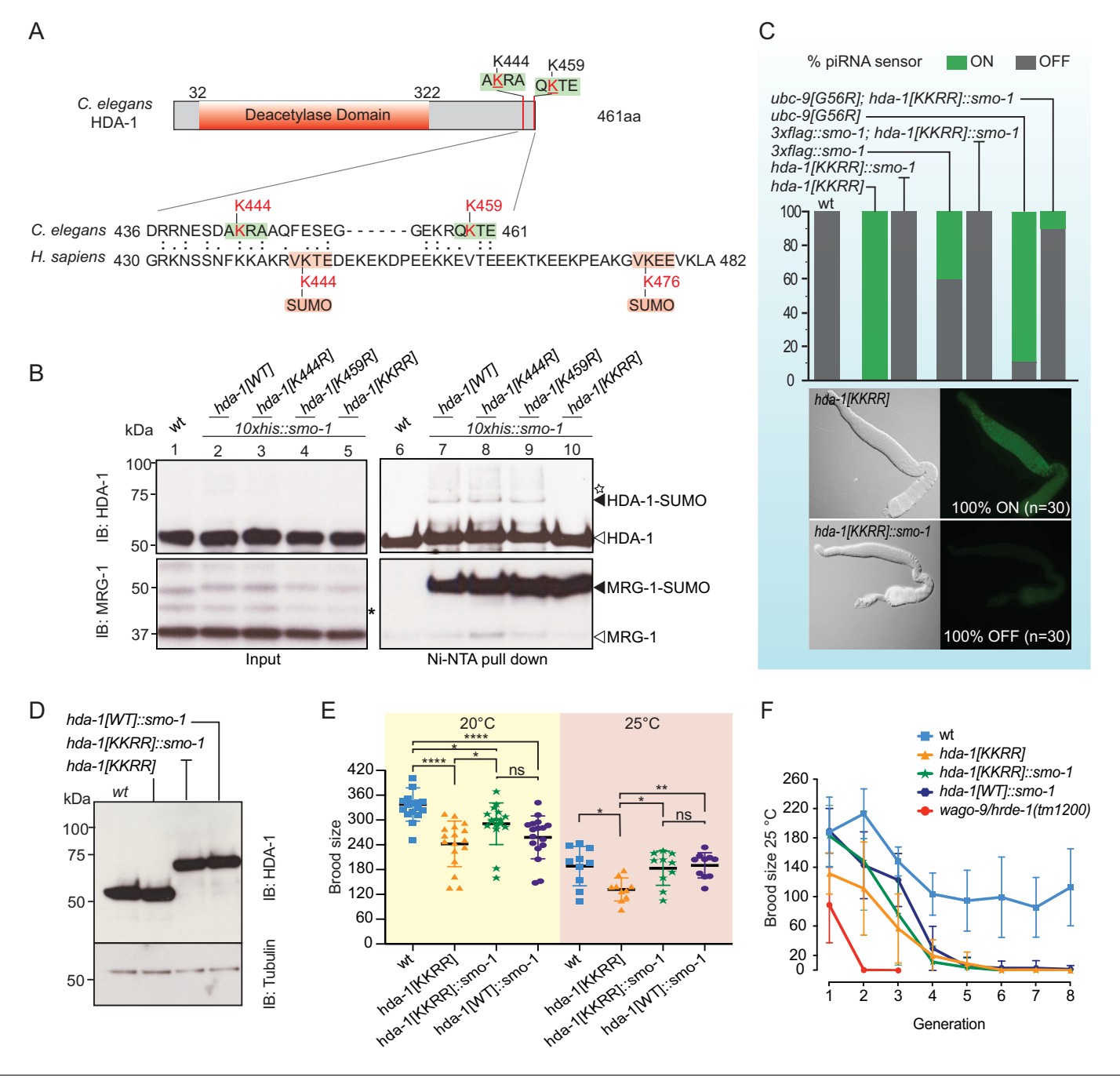

**Figure 2.** SUMOylation of HDA-1 at K444 and K459 facilitates piRNA-mediated silencing. (**A**) Domain structure of *C. elegans* type 1 histone deacetylase HDA-1 and C-terminal location of SUMO-acceptor sites. Sequence alignment showing poor conservation at the C-termini of *C. elegans* HDA-1 and *Homo sapiens* HDAC1. The human HDAC1 C-terminus possesses two consensus SUMO-acceptor sites, K444 and K476 (acceptor lysines in red; consensus SUMO acceptor motif in pink box). GPS-SUMO predicts two candidate non-consensus SUMOylation sites in HDA-1, both near the C-terminus, K444 and K459 (red lysines in green boxes). (**B**) Western blot analyses of HDA-1 and MRG-1 before (lanes 1–5) and after (lanes 6–10) affinity enrichment of SUMOylated proteins from wild-type (wt) or *hda-1* SUMO acceptor-site mutants. SUMOylated proteins were enriched from worms expressing HIS10::SMO-1. Black arrowheads indicate SUMOylated HDA-1 and MRG-1; white arrowheads indicate unmodified forms of HDA-1 and MRG-1. Asterisks indicate non-specific bands. Additional higher forms (indicated by white star) were detected, suggesting Multi-monoSUMOylation or PolySUMOylation of HDA-1. (**C**) Analysis of piRNA-mediated silencing in SUMOylation-defective mutants and rescue by HDA-1::SMO-1 translational fusion. (Top) The color of each bar indicates the percentage of worms in which the piRNA sensor was silent (OFF, gray) or expressed (ON, green). Thirty (n = 30) worms of each genotype were examined. (Bottom) Differential interference contrast and epifluorescence images of dissected gonads from *hda-1[KKRR]* and *hda-1[KKRR]::smo-1*. (**D**) Western blots showing levels of HDA-1 and variants proteins expressed from the endogenous *hda-1* locus. Tubulin

*Figure 2 continued on next page*

Figure 2 continued

was used as a loading control. (E) Brood size analysis of wt worms or HDA-1 SUMOylation-site mutants, and rescue by HDA-1::SMO-1 translational fusion. Worms were grown at 20°C or 25°C, as indicated. Statistical significance was determined by ordinary one-way ANOVA: *p<0.05; **p<0.01; ****p<0.0001; ns: not significant. (F) Mortal germline analyses of wt or HDA-1 SUMOylation-site mutant, and HDA-1::SMO-1 fusion worms. The *wago-9/hrde-1* mutant has a severe mortal germline phenotype. Worms were passaged at 25°C for eight generations, and the average number of progeny from 10 individuals (n = 10) was determined at each generation. Error bars represent standard error of the mean (SEM).

The online version of this article includes the following source data for figure 2:

**Source data 1.** Brood size and germ line mortality.

## HDA-1 SUMOylation mutants cause temperature-dependent reductions in fertility

Factors that promote genome integrity and epigenetic inheritance are required for germline immortality: their loss causes a mortal germline phenotype, whereby fertility declines in each generation and this decline is often exacerbated at elevated temperatures (*Ahmed and Hodgkin, 2000*). Although *hda-1* is an essential gene required for embryonic development (*Shi and Mello, 1998*), worms expressing HDA-1(KKRR) and HDA-1(KKRR)::SMO-1 from the endogenous *hda-1* locus were viable and fertile. Careful examination of brood size revealed that HDA-1(KKRR) worms made significantly fewer progeny than wild-type worms at 20°C and at 25°C (*Figure 2E*). When maintained at 25°C, the fertility of HDA-1(KKRR) worms steadily declined over several generations, from an average of 132 progeny in the first generation to fewer than 10 progeny in the fifth and subsequent generations (*Figure 2F*). Wild-type worms also showed an initial decline in fertility when maintained at 25°C, but averaged ~100 progeny in the fourth and subsequent generations (*Figure 2F*). By contrast, *wago-9/hrde-1* mutants showed a rapid decline in fertility and could not be maintained beyond the third generation (*Figure 2F*; *Buckley et al., 2012*; *Spracklin et al., 2017*). Appending SUMO via translational fusion, HDA-1(KKRR)::SMO-1, rescued the fertility defect of HDA-1(KKRR) worms, suggesting that HDA-1 SUMOylation promotes fertility (*Figure 2E*). When maintained at 25°C, however, HDA-1(KKRR)::SMO-1 worms gradually became infertile over five generations (*Figure 2F*). Worms expressing HDA-1::SMO-1—with intact SUMO-acceptor sites—were similar to HDA-1(KKRR)::SMO-1 animals, exhibiting significantly reduced fertility at 20°C and a further progressive decline in fertility over five generations at 25°C (*Figure 2F*). Thus, properly regulated SUMOylation of HDA-1 is essential for germline immortality.

## SUMOylation promotes HDA-1 association with other chromatin factors including NuRD complex components

SUMOylation modulates protein interactions (*Hendriks and Vertegaal, 2016*; *Kerscher, 2007*). To examine how SUMOylation affects HDA-1 complexes, we introduced a GFP tag into the C-terminus of the endogenous wild-type and mutant *hda-1* alleles and used GFP-binding protein (GBP) beads to immunoprecipitate the HDA-1::GFP fusion proteins (*Rothbauer et al., 2008*). SDS-PAGE analysis revealed that a core set of proteins strongly interact with HDA-1::GFP (*Figure 3A*). Mass spectrometry (MS) of the corresponding gel slices identified these proteins as: LIN-40, a homolog of metastasis-associated protein (MTA1) (*Chen and Han, 2001*); LIN-53, a homolog of retinoblastoma-associated protein 46/48 (RBAP46/48) (*Solari and Ahringer, 2000*); DCP-66, a homolog of GATA zinc finger domain containing protein GATAD (*Käser-Pébernard et al., 2014*); and SPR-5, a homolog of lysine demethylase (LSD1/KDM1) (*Katz et al., 2009*).

We also used reversed-phase high-performance liquid chromatography (RP-HPLC) MS to identify HDA-1::GFP interactors (see Materials and methods). Among approximately 200 high-confidence interactors with ≥10 spectral counts, we identified 63 proteins that were depleted by an arbitrary cutoff of 40% in immunoprecipitates from both *smo-1(RNAi)* and *hda-1[KKRR]::gfp* lysates (*Figure 3B, C* and *Supplementary file 2*). These SUMO-dependent interactors included MEP-1 (*Unhavaithaya et al., 2002*), AMA-1 (major subunit of pol II) (*Sanford et al., 1983*), and MET-2 (SETDB1 H3K9 histone methyltransferase) (*Andersen and Horvitz, 2007*; *Bessler et al., 2010*). This analysis also revealed that the core interactors identified above, LIN-40, LIN-53, DCP-66, and SPR-5, were reduced (by 12–37%) in immunoprecipitations (IPs) from *smo-1(RNAi)* or HDA-1(KKRR) lysates

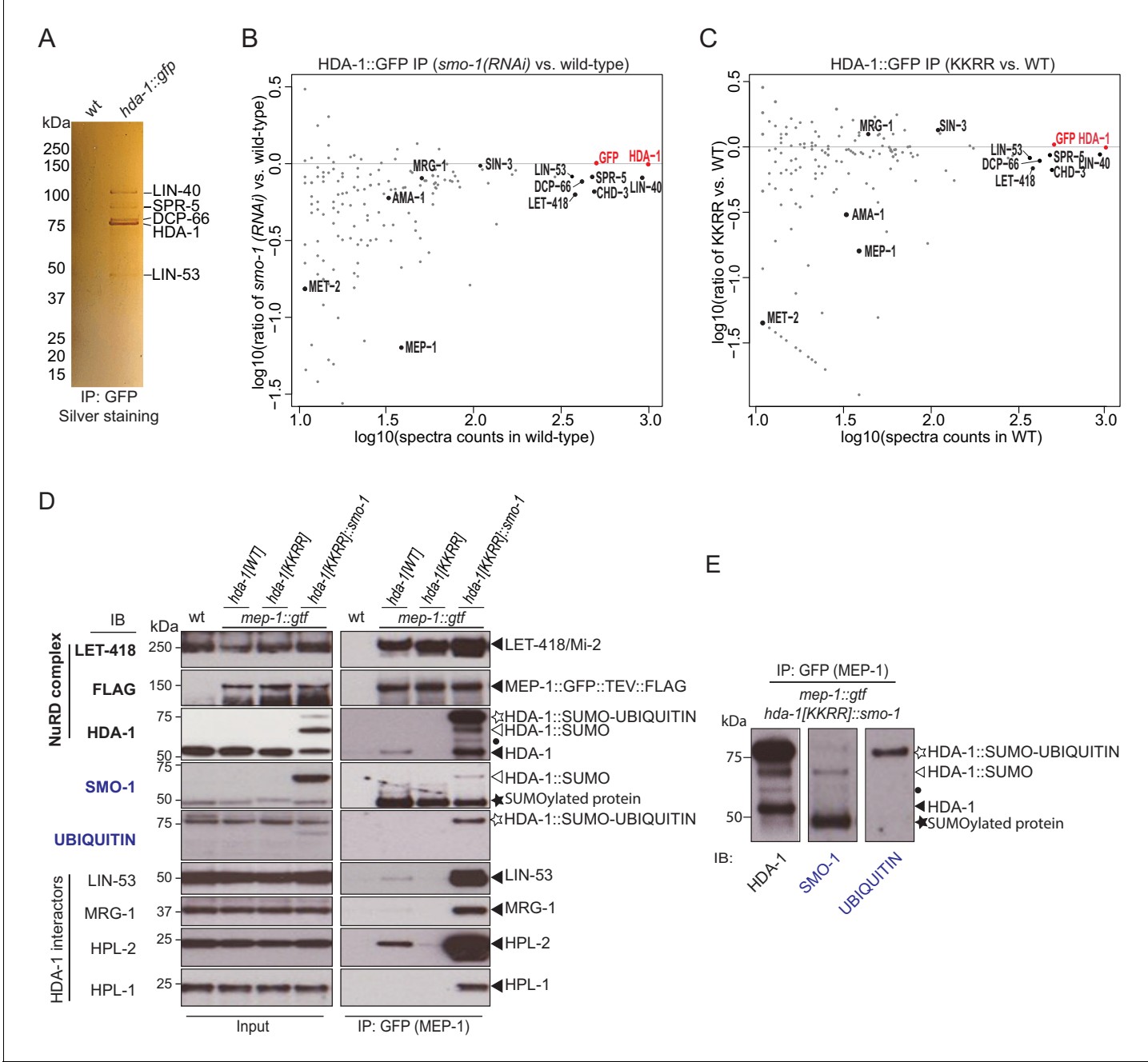

**Figure 3.** SUMOylation of HDA-1 promotes its association with NuRD and other chromatin factors. (**A**) Silver stained gel of proteins that co-immunoprecipitate with HDA-1::GFP. The indicated protein bands were excised from the gel and identified by mass spectrometry. (**B**) Scatter plot comparing the levels of proteins identified by mass spectrometry in HDA-1::GFP IPs from *smo-1(RNAi)* and wild-type (no RNAi) worms. The x axis shows the log value of spectral counts for each protein identified by IP-MS from wild-type worms. The y axis shows the log ratio of spectral counts for each protein in HDA-1::GFP IPs from *smo-1(RNAi)* vs. wild-type. (**C**) As in B, but comparing *hda-1[KKRR]::gfp* to *hda-1[WT]::gfp*. In (**B**) and (**C**), the spectral counts of HDA-1 (RED) were used to normalize between samples. A full list of the identified proteins is provided in *Supplementary file 2*. (**D**) Western blot analyses of proteins (indicated to left of blots) that associate with MEP-1::GTF (GFP IP) in *hda-1[WT]*, *hda-1[KKRR]* and *hda-1[KKRR]::smo-1* lysates. The detected proteins are indicated to the right (black arrowheads). The modified isoforms, HDA-1::SUMO and HDA-1::SUMO-UBIQUITIN are indicated with white arrow and white star, respectively. (**E**) Side-by-side comparison of HDA-1 isoforms detected in the HDA-1, SMO-1, and UBIQUITIN blots in (**D**). The black dot indicates an unknown HDA-1 isoform. The black star indicates an unknown SUMOylated protein.

(*Figure 3B, C* and *Supplementary file 2*). Of note, interactions between HDA-1 and SIN-3/SIN3A were not sensitive to SUMO-pathway perturbations (*Figure 3B, C* and *Supplementary file 2*).

## HDA-1 SUMOylation promotes the association of MEP-1 with chromatin regulators

To explore the molecular consequences of HDA-1 SUMOylation on its physical interactions with MEP-1 and other chromatin regulators, we crossed *hda-1[KKRR]* and *hda-1[KKRR]::smo-1* strains to worms that express a tandemly tagged MEP-1::GFP::TEV::3XFLAG (MEP-1::GTF) from the endogenous *mep-1* locus (*Kim et al., 2021*). We then used GBP beads to immunoprecipitate MEP-1::GTF protein complexes from *hda-1* wild-type or mutant lysates. As expected, interactions between MEP-1::GTF and LET-418/Mi-2 did not depend on HDA-1 SUMOylation (*Figure 3D*; *Kim et al., 2021*). Consistent with our HDA-1 proteomics studies, however, MEP-1::GTF pulled down wild-type HDA-1 but not HDA-1(KKRR) (*Figure 3D*). Moreover, LIN-53, MRG-1, and the HP-like heterochromatin proteins HPL-1 and HPL-2 were also greatly reduced in MEP-1 complexes purified from *hda-1[KKRR]* lysates (*Figure 3D*). Strikingly, each of these factors were dramatically increased, above wild-type levels, in MEP-1 complexes purified from *hda-1[KKRR]::smo-1* lysates (*Figure 3D*). Thus, the C-terminal fusion of SUMO to HDA-1 rescues MEP-1 interactions with HDA-1(KKRR) and also promotes MEP-1 interaction with HDA-1-binding partners. It is important to note that SUMO-conjugation (in contrast to the SUMO-translational fusion) is rapidly reversed in lysates prepared for IP studies (*Kim et al., 2021*), which explains why the bands detected by western blotting in *Figure 3D*—even for the strongly SUMOylated MRG-1—migrate at the size of the unmodified proteins.

In MEP-1 immunoprecipitates from *hda-1[KKRR]::smo-1* lysates, we detected multiple HDA-1 bands, including a prominent band slightly larger than the expected size of HDA-1::SMO-1 (white stars in *Figure 3D, E*). Western blots with ubiquitin-specific antibody suggested that this prominent band is a mono-ubiquitinated form of the fusion protein (*Figure 3D, E*). In both input and IP samples, we also observed an isoform similar in size to endogenous HDA-1 that was not detected by SUMO-specific antibodies (*Figure 3D, E*), suggesting that the HDA-1::SMO-1 fusion protein may be cleaved near the C-terminus of HDA-1, removing the SUMO peptide.

## HDA-1 SUMOylation promotes histone deacetylation in vivo

The findings above suggest that SUMOylation promotes the assembly and function of HDA-1 complexes. Our proteomic studies also revealed that SUMOylation promotes interactions between HDA-1 and other histone-modifying enzymes required for heterochromatin formation, including the demethylase SPR-5/LSD1, which removes activating H3K4me2/3 marks, and the methyltransferase MET-2/SetDB1, which installs silencing H3K9me2/3 marks (*Greer and Shi, 2012*). Consistent with the idea that SUMOylation of HDA-1 promotes silencing via SPR-5 and MET-2, immunostaining revealed greatly reduced levels of H3K9me2 and increased levels of the H3K4me3 and throughout the germline in HDA-1(KKRR) worms as compared to wild-type (*Figure 4A, B*, *Figure 4—figure supplement 1*). Immunostaining also revealed higher levels of acetylated H3K9 (H3K9Ac) in germlines of HDA-1(KKRR) and *mep-1*-depleted worms than in wild-type (*Figure 4C*). Moreover, we found that HDA-1, LET-418, and MEP-1 (including MEP-1 expressed from the germline-specific *wago-1* promoter) bind heterochromatic regions of the genome, depleted of the activating H3K9Ac mark and enriched for the silencing marks H3K9me2/3 (*Figure 4D*). Thus, SUMOylation of HDA-1 appears to drive formation or maintenance of germline heterochromatin.

## SUMOylated HDA-1 and PRG-1 co-regulate hundreds of targets, including many spermatogenesis genes

The visibly reduced level of germline heterochromatin in HDA-1(KKRR) worms suggests that gene expression is broadly misregulated when HDA-1 cannot be SUMOylated. To examine the effect of HDA-1 SUMOylation on germline gene expression, we performed high-throughput sequencing of mRNAs isolated from dissected germlines of *hda-1[KKRR]*, *hda-1[KKRR]*::smo-1, and *ubc-9[G56R]* animals, and from worms expressing degron alleles of *hda-1* and *mep-1* with or without auxin exposure beginning at the L4 stage (*Figure 5A*). Replicate libraries gave highly reproducible mRNA profiles from each mutant (*Figure 5—figure supplement 1A*). Depletion of HDA-1 or MEP-1 or inactivation of the SUMO pathway caused widespread upregulation of germline mRNAs and transposon families,

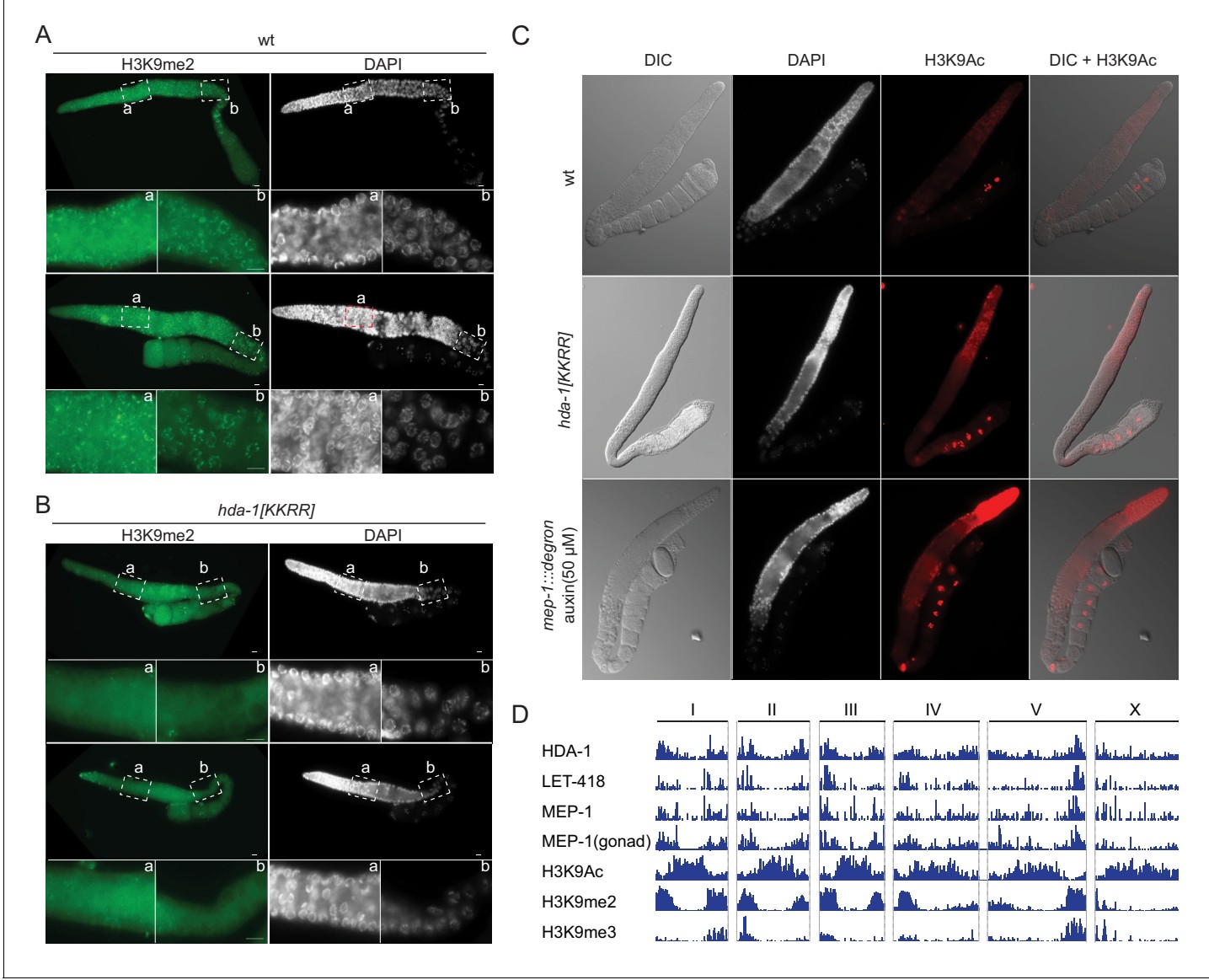

**Figure 4.** HDA-1 SUMOylation is required for formation of germline heterochromatin. (**A, B**) Immunofluorescence micrographs of (**A**) wild-type (wt) and (**B**) *hda-1[KKRR]* gonads stained with anti-H3K9me2 antibody and DAPI. The dashed boxes indicated 'a' and 'b' are enlarged as shown. Two representative gonads are shown for each strain. (**C**) Differential interference contrast and immunofluorescence micrographs of gonads from wt, *hda-1 [KKRR]*, *mep-1::gfp::degron* with auxin (50 µM) worms stained with anti-H3K9Ac antibody and DAPI. (**D**) Genome Browser tracks (Integrated Genomics Viewer [IGV]) showing ChIP-seq peaks for NuRD complex components (HDA-1, LET-418, and MEP-1) and three histone modifications (H3K9Ac, H3K9me2, and H3K9me3) along each *C. elegans* chromosome (I–V and X). MEP-1(gonad) data are from worms that express MEP-1::GTF only in the germline, using the *wago-1* promoter, for germline-specific CHIP.

The online version of this article includes the following figure supplement(s) for figure 4:

**Figure supplement 1.** Increased levels of active H3K4me3 chromatin mark in *hda-1[KKRR]*.

with extensive but incomplete overlap between the mutants (*Figure 5B, Figure 5—figure supplement 1B, Figure 5—figure supplement 2*). Twice as many genes were upregulated in *degron::hda-1* germline as in *degron::mep-1* or *ubc-9[G56R]* (*Figure 5B, Figure 5—figure supplement 1B*), likely reflecting the role of HDA-1 in multiple complexes. Nearly 10-fold fewer genes were upregulated in *hda-1[KKRR]* than in auxin-treated *degron::hda-1* animals (*Figure 5B*, *Figure 5—figure supplement 1B*), consistent with the phenotypic differences between the two mutants. Most (305, ~71%) of the genes upregulated in *hda-1[KKRR]* germlines were also upregulated in auxin-treated *degron::hda-1*

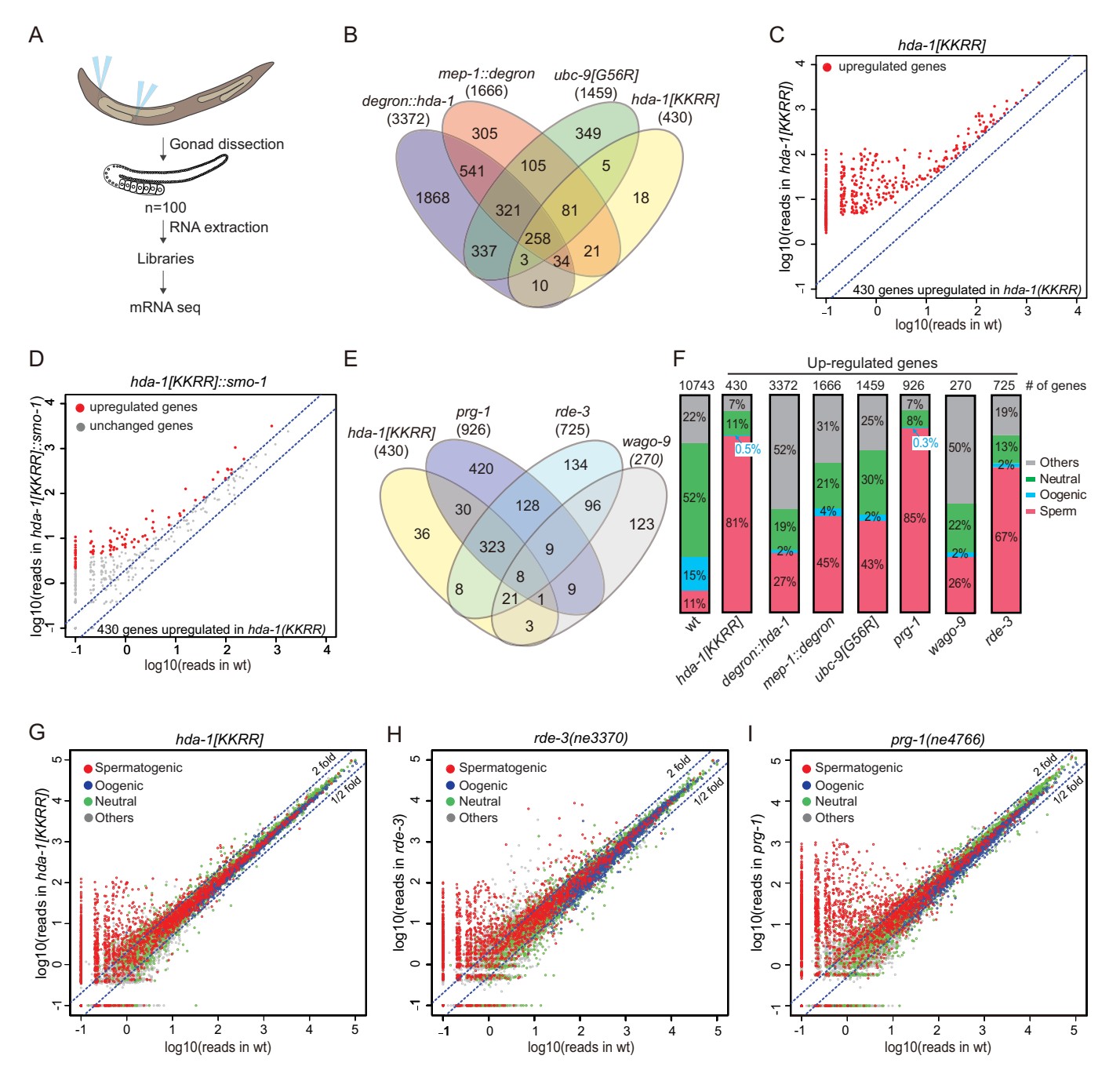

**Figure 5.** The SUMO, NuRD, and piRNA pathways regulate the same group of targets. (**A**) Schematic of mRNA-seq from dissected gonads. (**B**) Venn diagram showing overlap between upregulated genes in *degron::hda-1*, *mep-1::degron*, *ubc-9(G56R)*, and *hda-1[KKRR]* germlines. Numbers in parentheses indicate total number of upregulated genes. (**C**) Scatter plot of upregulated genes in *hda-1[KKRR]*. The x-axis represents reads in wild-type (wt), and y-axis represents reads in *hda-1[KKRR]*. (**D**) Scatter plot showing the effect of *hda-1[KKRR]::smo-1* on the 430 genes (from **C**) upregulated in *hda-1[KKRR]*. The x-axis represents reads in wt, and y-axis represents reads in *hda-1[KKRR]::smo-1*. (**E**) Venn diagram showing overlap between upregulated genes in *hda-1[KKRR]*, *prg-1*, *rde-3*, and *wago-9* mutants. (**F**) Bar graph showing fractions of upregulated genes involved in spermatogenesis, oogenesis, neutral, or other categories. 'Other' indicates genes that cannot be put into one of the other categories (*Ortiz et al., 2014*). Genes with >1 mRNA-seq reads in wt gonad were used to generate the 'wild-type' dataset as a reference. The number of genes in each dataset is labeled at the top. (**G–I**) Scatter plots comparing mRNA-seq reads in (**G**) *hda-1[KKRR]*, (**H**) *rde-3*, and (**I**) *prg-1(ne4766)* to those in wt. The blue dashed lines indicate a twofold increase or decrease in mutant compared to wt.

The online version of this article includes the following figure supplement(s) for figure 5:

*Figure 5 continued on next page*

*Figure 5 continued*

**Figure supplement 1.** Comparison of replicates in gonad mRNA-seq.
**Figure supplement 2.** Analysis of transposons.
**Figure supplement 3.** Loss of HDA-1 SUMOylation leads to depletion of H3k9me2 on a selected set of genes.
**Figure supplement 4.** Upregulated spermatogenic genes in the mutants.
**Figure supplement 5.** Analysis of small RNAs targeting upregulated genes in *hda-1[KKRR]*.

animals (*Figure 5B*). As expected, mRNAs upregulated in *hda-1[KKRR]* were restored to nearly wild-type levels in *hda-1[KKRR]::smo-1* (*Figure 5C, D*). Consistent with the known roles of the NuRD complex and SUMOylation pathways in modulating chromatin states, we observed a loss of enrichment for H3K9me2, as measured by ChIP-seq, near the promoters of genes upregulated in *hda-1[KKRR]*, *ubc-9[G56R]*, and auxin-treated *degron::hda-1* worms (*Figure 5—figure supplement 3*).

Because HDA-1 SUMOylation is required for silencing of a piRNA reporter, we examined which endogenous mRNAs are co-regulated by HDA-1 SUMOylation and the piRNA pathway factors, PRG-1, WAGO-9/HRDE-1, and RDE-3. We prepared mRNA sequencing libraries from dissected gonads collected from *prg-1(ne4766)*, *wago-9/hrde-1(tm1200)*, and *rde-3(ne3370)* mutant animals (*Figure 5A*). RDE-3 is required for production of small RNAs (termed 22G-RNAs) that guide transcriptional and post-transcriptional silencing by WAGO Argonautes, including WAGO-9 (*Gu et al., 2009*; *Zhang et al., 2011*). Of the 430 mRNAs upregulated (≥2-fold) in *hda-1[KKRR]*, we found that 362 (84%) were upregulated (≥2-fold) in *prg-1(ne4766)*, 360 (84%) were upregulated in *rde-3 (ne3370)*, and 331 (77%) were upregulated in all three mutant strains (*Figure 5E*). Similarly, the genes upregulated in *hda-1[KKRR]* accounted for 40% of the genes upregulated in *prg-1(ne4766)* and 50% of those upregulated in *rde-3(ne3370)* (*Figure 5E*). Fewer genes were upregulated in *wago-9/hrde-1* (*Figure 5E*, *Figure 5—figure supplement 1B*), perhaps due to redundancy with other WAGO Argonautes. Whereas only 5 transposon families were upregulated in *prg-1(ne4766)* and a total of 6 transposon families were upregulated in *hda-1[KKRR]* (*Figure 5—figure supplement 2*), thirty (30) were upregulated in *rde-3* mutants (*Figure 5—figure supplement 2*), consistent with the previously described role for *rde-3* and the WAGO pathway in maintaining the silencing of most transposons in worms. The silencing defect was more severe in auxin-treated *degron::hda-1* than in *hda-1[KKRR]* (*Figure 5—figure supplement 1B, Figure 5—figure supplement 2*), resulting in the increased expression of many more transposons and a more extensive overlap with genes upregulated in *rde-3* mutant worms (*Figure 5—figure supplement 1C, Figure 5—figure supplement 2*). This result indicates that HDA-1 also promotes the maintenance of silencing independently of HDA-1 SUMOylation.

Most of the genes upregulated in *prg-1*, *rde-3*, and *hda-1[KKRR]* mutants are normally expressed during spermatogenesis (*Figure 5F–I*, *Figure 5—figure supplement 4*). In most cases, the upregulation of these spermatogenesis mRNAs did not correlate with reduced WAGO 22G-RNAs targeting these genes (*Figure 5—figure supplement 5*), suggesting that the spermatogenesis switch may be regulated indirectly by the small RNA pathways.

## HDA-1 physically interacts with WAGO-9/HRDE-1 and functions in inherited RNAi

Because WAGO-9/HRDE-1 is a nuclear Argonaute that functions downstream in the Piwi pathway to establish and maintain epigenetic silencing (*Ashe et al., 2012*; *Bagijn et al., 2012*; *Buckley et al., 2012*; *Shirayama et al., 2012*), we asked if WAGO-9/HRDE-1 interacts with HDA-1. We used CRISPR genome editing to generate a functional *gfp::wago-9* strain and then used GBP beads to immunoprecipitate GFP::WAGO-9 complexes. Western blot analyses revealed that HDA-1 co-precipitates specifically with GFP::WAGO-9 (*Figure 6A*). We also found that the HP1-like protein HPL-2 interacts with WAGO-9/HRDE-1 (*Figure 6A*), consistent with previous genetic studies (*Ashe et al., 2012*; *Buckley et al., 2012*; *Gu et al., 2012*; *Luteijn et al., 2012*; *Shirayama et al., 2012*). HPL-2 binds methylated H3K9 in heterochromatin (*Garrigues et al., 2015*). Neither HDA-1 nor HPL-2 were precipitated by GBP beads incubated with lysates prepared from untagged wild-type worms. These interactions were confirmed by reciprocal GFP IP experiments using *hda-1::gfp* lysates (*Figure 6B*). Moreover, the interaction of HDA-1 with WAGO-9 and HPL-2 was reduced in *hda-1[KKRR]* animals

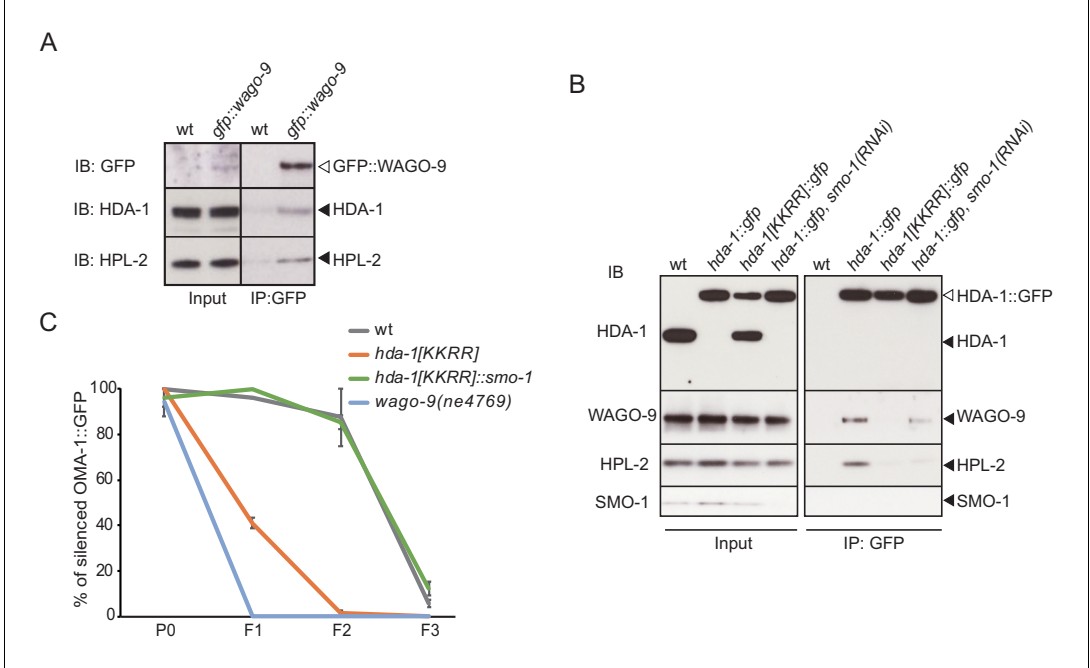

**Figure 6.** HDA-1 interacts with WAGO-9/HRDE-1, which requires HDA-1 SUMOylation. (**A**) Western blot analysis showing that HDA-1 and HPL-2 co-immunoprecipitate with GFP::WAGO-9/HRDE-1. (**B**) Western blot analysis of WAGO-9 and HPL-2 in HDA-1::GFP immunoprecipitates from wild-type, HDA-1 SUMO acceptor-site mutant, or *smo-1(RNAi)* worms. Blotting with anti-SMO-1 antibody showed depletion of SMO-1 in the *smo-1(RNAi)* worms. A band the size of untagged HDA-1 (black arrowhead) in *hda-1[KKRR]::gfp* in input appears to be a cleavage product that removes the GFP tag. (**C**) Graph showing the levels of silencing induced by RNAi over three generations in wild-type, *hda-1[KKRR]*, and *hda-1[KKRR]::smo-1* worms. Worms were treated with *gfp(RNAi)* at P0, and F1 larvae were transferred to regular NGM plates. The percentage of worms that express OMA-1::GFP was scored (n≥22, two replicates). Error bars represent standard error of the mean (SEM).

or by *smo-1(RNAi)* (*Figure 6B*). Interestingly, we observed a truncated form of HDA-1(KKRR)::GFP in the input samples that appears to result from proteolytic removal of the C-terminal GFP (*Figure 6B*, filled arrow, compare input to IP lanes). By contrast, GFP is not proteolytically removed from wild-type HDA-1::GFP in the presence or absence of *smo-1(RNAi)*. The reason for this difference in protein stability is not known and will require further study. Taken together, the SUMO-dependent interactions between HDA-1 and WAGO-9 suggest that SUMOylation of HDA-1 promotes the assembly of an Argonaute-guided nucleosome-remodeling complex.

Many of the factors that maintain heritable epigenetic silencing triggered by piRNAs—including WAGO-9/HRDE-1—are also required for multigenerational germline silencing triggered by exogenous dsRNA, that is, inherited RNAi (*Ashe et al., 2012*; *Buckley et al., 2012*; *Shirayama et al., 2012*). We therefore asked if HDA-1 SUMOylation is also required for inherited RNAi. Worms expressing a bright germline OMA-1::GFP were fed bacteria that express GFP dsRNA (i.e., RNAi by feeding) for one generation (P0). The subsequent F1 and F2 generations were removed from the RNAi food and cultured on a normal *Escherichia coli* diet (no *gfp* dsRNA). In wild-type worms, *oma-1::gfp* was silenced in the P0 generation (in the presence of *gfp* dsRNA) and remained silent in the F1 and F2 generations in the complete absence of *gfp* dsRNA (*Figure 6C*). By contrast, as previously shown (*Ashe et al., 2012*; *Buckley et al., 2012*; *Spracklin et al., 2017*), *oma-1::gfp* is efficiently silenced in P0 *wago-9/hrde-1* worms in the presence of *gfp* dsRNA, but *oma-1::gfp* is reactivated in the F1 and F2 generation after the removal from *gfp* dsRNA. *hda-1[KKRR]* animals were similarly defective for heritable RNAi: *oma-1::gfp* was efficiently silenced in the P0 worms in the presence of *gfp* dsRNA, but expression was fully restored in the F1 and F2 generations in the absence of *gfp* dsRNA (*Figure 6C*). Finally, the *hda-1[KKRR]::smo-1* translational fusion fully rescued the heritable RNAi defect of *hda-1[KKRR]* (*Figure 6C*). Taken together, these findings suggest that HDA-1 SUMOylation promotes heritable transcriptional silencing in both the piRNA and RNAi pathways.

## Discussion

### HDAC1 SUMOylation promotes Argonaute-directed transcriptional silencing

Argonaute small RNA pathways collaborate with chromatin factors to co-regulate gene expression, transposon activity, and chromosome dynamics (reviewed in *Almeida et al., 2019*). Here, we have identified components of the NuRD complex as well as the SUMO and its E2 protein ligase, UBC-9, as chromatin-remodeling and -modifying complexes required for the initiation of Piwi-mediated gene silencing. Using mutagenesis of candidate SUMO-acceptor sites and affinity enrichment of SUMO, we identified C-terminal lysines required for the SUMOylation of the HDAC1 homolog, HDA-1. Mutating these lysines to arginine in HDA-1(KKRR) abolished the initiation but not the maintenance of piRNA-mediated silencing and also abolished the initiation of transgenerational silencing in response to dsRNA. Appending SUMO by translational fusion to HDA-1(KKRR) completely rescued Argonaute-mediated silencing and even restored piRNA silencing in animals with mutations in the SUMO conjugating machinery. Taken together, these molecular genetic studies strongly implicate HDAC1 SUMOylation in promoting Argonaute-directed transcriptional silencing (*Figure 7*).

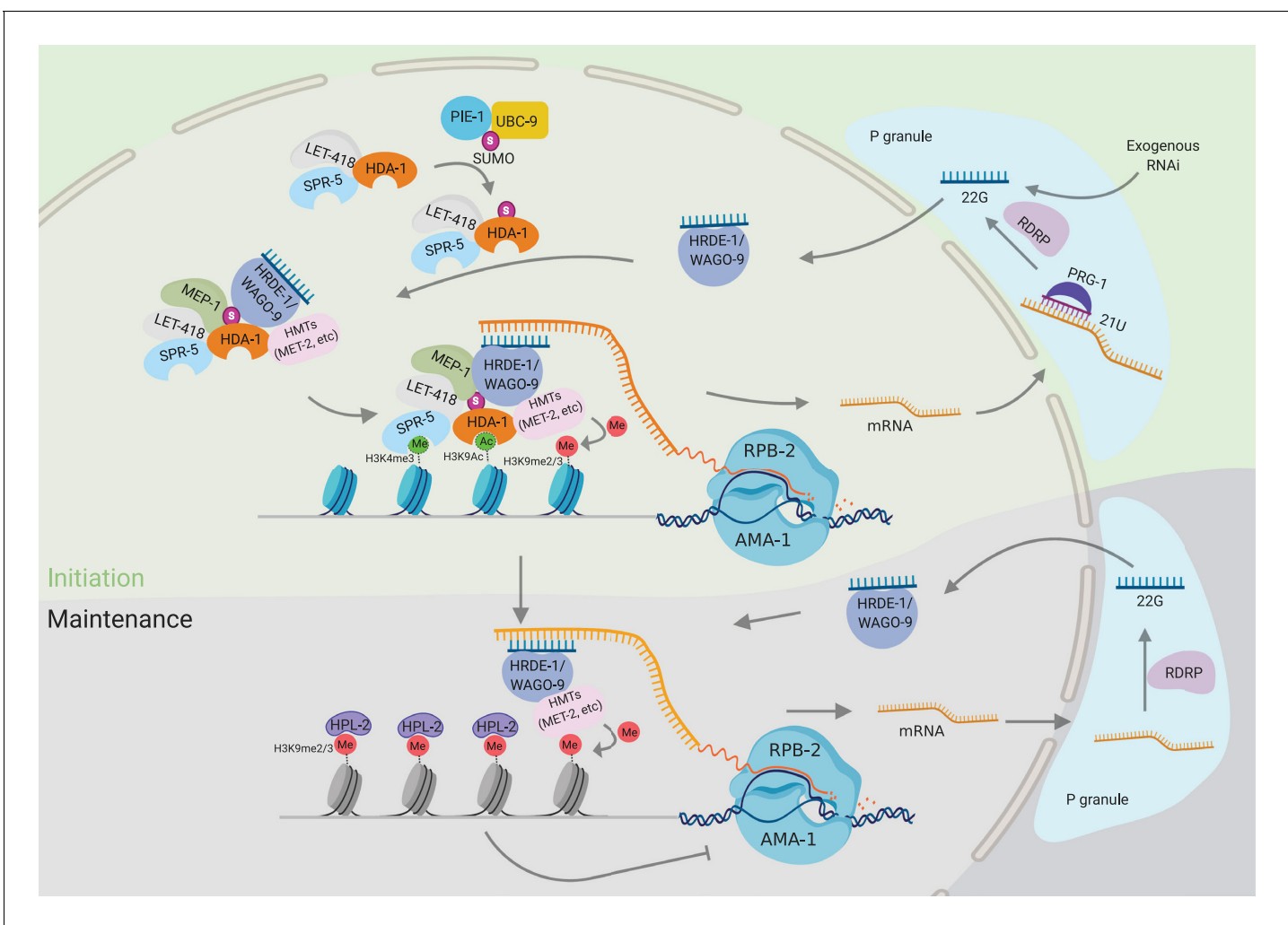

**Figure 7.** Model: HDAC1 SUMOylation promotes Argonaute-directed transcriptional gene silencing. SUMOylation of HDA-1 enables nucleosome remodeling and deacetylase (NuRD) complex assembly in the adult germline and WAGO-9 or other Argonautes recruit the NuRD complex to piRNA targets. The Argonaute/NuRD complex, along with other histone-modifying enzymes—for example, SPR-5, MET-2—removes the active histone marks (H3K9Ac and H3K4me2/3) and establishes silencing marks (H3K9me2/3) to suppress their transcription.

## SUMO: a potent genetic modifier with an elusive biochemical signature

Given the strong genetic evidence that the modification of HDA-1 by SUMO promotes piRNA silencing in the adult germline, we were surprised that the conjugated isoform of HDA-1 was undetectable in our IP assays from adult animals. Only when Ni-Affinity purification was used under strict denaturing conditions was it possible to recover the HDA-1-SUMO isoform. A likely explanation for these findings is that the covalent attachment of SUMO to its target proteins is rapidly reversed in lysates from *C. elegans* adults (*Kim et al., 2021*).

The dynamic nature of SUMO conjugation is well known from studies on other organisms (*Mukhopadhyay and Dasso, 2007*), and the SUMO protease SENP1 has been identified as an important regulator of human HDAC1 SUMOylation (*Cheng et al., 2004*, see more discussion below). In the future, it will be interesting to explore whether any of the three *C. elegans* homologs of SENP1 regulate HDA-1 SUMOylation in the worm germline. The presence of such factors, which appear to be very active in worm lysates, likely explains why so little HDA-1-SUMO was recovered in our IP studies. Indeed, it seems likely that the labile nature of the SUMO interaction could explain the discordance between its robust genetic effect and its surprisingly weak physical association with its targets in co-IP assays. Consistent with this idea, an HDA-1::SMO-1 translational fusion protein with an abnormal linkage via the N-terminus of SUMO was stable in protein lysates, strongly rescued the silencing defects of the presumptive SUMO-acceptor mutant protein HDA-1(KKRR), and dramatically enhanced the detection of protein-protein interactions between HDA-1 and components of an adult-stage NuRD complex.

We were surprised that the HDA-1::SMO-1 fusion protein, which was constructed at the endogenous and essential *hda-1* locus, was so well-tolerated. Conceivably, the cleavage and ubiquitination of HDA-1::SMO-1 that we detected in our IP assays provide alternative mechanisms to counter the effects of this translational fusion. The mono-ubiquitinated form of the HDA-1::SMO-1 protein exhibited reduced staining with a SUMO-specific antibody, raising the possibility that the SUMO moiety in this protein is itself modified by ubiquitin, a modification that has been reported for SUMO isoforms in humans (*Danielsen et al., 2011*; *Tatham et al., 2008*). Taken together, our findings suggest that HDA-1 SUMOylation is not only important genetically but is highly dynamic and may be regulated at multiple levels.

## HDAC1 SUMOylation plays diverse roles in gene regulation

Studies in mice and in human cell culture have investigated SUMOylation of C-terminal HDAC1 lysines (*Cheng et al., 2004*; *Citro et al., 2013*; *David et al., 2002*; *Joung et al., 2018*; *Tao et al., 2017*). In at least three cases, mutating these lysines increased the levels of histone acetylation and transcriptional activation (*David et al., 2002*; *Cheng et al., 2004*; *Joung et al., 2018*). As mentioned above, the SUMO protease SENP1 was identified as a factor that promotes the removal of C-terminal SUMO moieties from human HDAC1 in a prostate cancer model, causing upregulation of the androgen receptor (*Cheng et al., 2004*). In a subsequent study, these authors identified SENP1 as upregulated in human prostate cancers and found that overexpression of SENP1 was sufficient to drive prostate neoplasia in a mouse model (*Cheng et al., 2006*). One very interesting study showed that Aβ insult led to upregulation of the SUMO E3 factor PIAS and resulted in SUMOylation of C-terminal lysines in HDAC1 in the rat hippocampus (*Tao et al., 2017*). Remarkably, this study showed that a lentivirus-driven HDAC1::SUMO translational fusion protein could rescue the learning and memory deficits of an APP/Presenilin 1 murine model of Alzheimer's disease (*Tao et al., 2017*), suggesting that HDAC1 SUMOylation may be neuroprotective in response to Aβ accumulation. Taken together, our worm studies and these studies in mammalian systems point to the likely importance of HDAC1 SUMOylation in the regulation of gene expression in diverse animals.

## SUMO promotes the assembly of a germline MEP-1-NuRD complex

Precisely how HDA-1 SUMOylation promotes its downstream functions in piRNA silencing will require further work. One possibility is that SUMO acts as a bridge between HDA-1 and a nuclear Argonaute WAGO-9, while simultaneously promoting NuRD complex assembly (*Figure 7* model). This model is supported by the dramatically enhanced association between MEP-1 and the HDA-1::SMO-1 translational fusion and by the ability of HDA-1::SMO-1 to rescue the piRNA silencing defects of upstream SUMO pathway mutants.

Paradoxically, HDA-1 is not SUMOylated in the embryo and yet interacts robustly with MEP-1 (*Kim et al., 2021*). We do not know why the association of HDA-1 with MEP-1 requires SUMO in the adult germline but not the embryo. Perhaps unknown co-factors promote their SUMO-independent interaction in the embryo. Alternatively, adult germline-specific factors may inhibit or modify HDA-1 or other NuRD complex components to prevent their direct association. SUMOylation of HDA-1 in the adult may therefore enable it to associate with MEP-1 via a mechanism bridged by SUMO, a SUMO glue, or SUMO-SIM network mode of interaction (*Matunis et al., 2006*; *Psakhye and Jentsch, 2012 Pelisch et al., 2017*). The MEP-1 protein has two consensus SIMs, which could be important for this association between MEP-1 and HDA-1. However, our preliminary studies on these motifs were confounded by the finding that their simultaneous mutation resulted in a completely sterile inactive *mep-1* allele, similar phenotypically to a null allele. Thus, although the SIM motifs in MEP-1 may be important for its interaction with HDA-1-SUMO, they may also be required for other functions or for interactions with other essential co-factors.

## Parallels in the role of SUMOylation in Piwi silencing in insects, mice, and worms

Histone deacetylation is a necessary step in de novo transcriptional silencing. Yet, precisely how nuclear Argonautes orchestrate both deacetylation and the subsequent installation of silencing marks on target chromatin is not known. The nuclear Argonaute WAGO-9/HRDE-1 initiates transcriptional silencing downstream of both the piRNA- and the dsRNA-induced silencing pathways (*Ashe et al., 2012*; *Bagijn et al., 2012*; *Buckley et al., 2012*; *Shirayama et al., 2012*), and we have shown here that HDA-1 interacts with WAGO-9 in a manner that partially depends on HDA-1 SUMOylation. In fruit fly, the nuclear Argonaute Piwi was recently shown to interact with the SUMO E3 PIAS homolog Su(var)2–10 and to promote the recruitment of the histone methyltransferase SetDB1/EGGless (Egg) and its cofactor MCAF1/Windei (Wde) (*Ninova et al., 2020*). A worm paralog of EGG, MET-2, was identified in our SUMO-dependent HDA-1 IP complexes. Although the worm PIAS/Su(var)2–10 homolog, GEI-17, scored negative in our piRNA sensor screen, in a parallel study we have identified a synthetic piRNA silencing defect between *gei-17* and mutations in *pie-1*, which encodes a tandem zinc finger protein with properties consistent with SUMO E3 activity (*Kim et al., 2021*). Thus, an attractive possibility is that upon binding to nascent target transcripts, nuclear Argonautes recruit SUMO E3 factors to promote the SUMOylation and recruitment of both histone deacetylase and histone methyltransferase complexes.

Interestingly, in *Drosophila* Kc167 cells, a MEP-1:Mi-2 complex (called MEC) reportedly forms without HDAC1 (*Kunert et al., 2009*). Kc167 cells are derived from ovarian somatic cells and express components of the Piwi pathway (*Vrettos et al., 2017*), raising the possibility that HDAC1 SUMOylation might promote its association with the MEC complex in these cells. Interestingly, a recent paper showed that *Drosophila* MEP-1 and the HDA-1 homolog RPD3 interact with PIWI and with Su(var)2–10 in fly ovarian somatic cells, where they function together to promote transposon silencing (*Mugat et al., 2020*). Another recent study found that the mouse Piwi homolog MIWI engages NuRD complex components and DNA methyltransferases to establish de novo silencing of transposons in the mouse testes, suggesting additional parallels to the findings from worms and flies (*Zoch et al., 2020*). It will be interesting in the future to learn if mammalian and insect Piwi Argonautes target HDAC1 SUMOylation to promote de novo Piwi silencing.

The *mep-1* gene was previously implicated in regulating the transition from spermatogenesis to oogenesis in hermaphrodites (*Belfiore et al., 2002*), and our findings suggest that HDA-1 SUMOylation also promotes this function. We were surprised, however, to find that spermatogenesis targets are also regulated by the WAGO-pathway factor RDE-3 and by the Piwi Argonaute PRG-1. Similar findings were reported recently by *Reed et al., 2020*. These observations raise the interesting possibility that *C. elegans* germline, Argonaute systems function with SUMO and HDAC1 in promoting the switch from spermatogenesis to oogenesis in hermaphrodites.

## Materials and methods

### *C. elegans* strains and genetics

All the strains in this study were derived from Bristol N2 and cultured on nematode growth media (NGM) plates with OP50, and genetic analyses were performed essentially as described (*Brenner, 1974*). The strains used in this study are listed in *Supplementary file 3*.

### RNAi screen

RNAi screen was performed against all 337 genes in the chromatin subset of the *C. elegans* RNAi collection (Ahringer). RNAi of *smo-1* and *ubc-9* were added to the screen as chromatin regulators. Synchronous L1 worms of the reporter strain were plated on the isopropyl-β-D-thiogalactoside (IPTG) plates with the corresponding RNAi food. Bacteria with empty L4440 vector served as negative control. The desilencing phenotype was scored when the worms on the control plates grew to young adult stage at 20℃. In the first round of the screen, 78 RNAi clones scored positive. In the second round, we required that the sensor be desilenced in more than 5% of worms, resulting in 29 positive clones, including *smo-1* and *ubc-9* (*Supplementary file 1*).

### CRISPR/Cas9 genome editing

The Co-CRISPR strategy (*Kim et al., 2014*) and *mep-1* sgRNA and *smo-1* sgRNA (see *Kim et al., 2021*) were used to generate *mep-1::gfp::degron* and *3xflag::smo-1* strains. Other CRISPR lines were generated by Cas9 ribonucleoprotein (RNP) editing (*Dokshin et al., 2018*) or Cas12a (cpf1) RNP editing. Cas12a genome editing mixture containing Cas12a protein (0.5 μl of 10 μg/μl), two crRNAs (each 2.8 μl of 0.2 μg/μl), annealed PCR donor (4 μg), and *rol-6(su1006)* plasmid (2 μl of 500 ng/μl) was incubated at 37℃ for 30 min and 25℃ for 1 hr or overnight before injecting animals. For short insertions, like FLAG, auxin-inducible degron, HIS10, and point mutations, synthetic single-strand DNAs were used as donors; for long insertions, like GFP, 2/3xFLAG-Degron, and SUMO, annealed PCR products were used as donors. Guide RNA sequences and ssOligo donors used in this study are listed in *Supplementary file 4*.

### Generation of strains expressing SUMO-conjugated HDA-1

To prevent transfer of SUMO from the HDA-1::SMO-1 fusion to other SUMO targets, we mutated the tandem C-terminal glycines of SMO-1/SUMO to alanines (GG to AA) (*Dorval and Fraser, 2006*). The modified *smo-1* open reading frame was fused directly before the stop codon of *hda-1* or *hda-1 (ne4747[KKRR])* gene at the endogenous *hda-1* locus by CRISPR genome editing as described above.

### Auxin treatment

For the auxin-inducible degron system (*Zhang et al., 2015*), *tir-1::mRuby* was expressed in the germline under the control of the *sun-1* promoter and *eft-3* 3′ UTR. The degron-tagged L1 larvae were plated on NGM plates with 100 μM auxin indole-3-acetic acid (IAA; Alfa Aesar, A10556) unless otherwise stated and kept in dark. Worms were collected at young adult stage for further analysis.

### Gonad fluorescent image

Gonads were dissected on glass slide (Thermo Fisher Scientific, 1256820) in M9 buffer, mounted in 2% paraformaldehyde (Electron Microscopy Science, Nm15710) in egg buffer (25 mM HEPES pH 7.5, 118 mM NaCl, 48 mM KCl, 2 mM CaCl$_2$, 2 mM MgCl$_2$), and directly imaged. Epifluorescence and differential interference contrast (DIC) microscopy were performed using an Axio Imager M2 Microscope (Zeiss). Images were captured with an ORCA-ER digital camera (Hamamatsu) and processed using Axiovision software (Zeiss).

### Immunofluorescence

Immunostaining of gonads was performed essentially as described (*Kim et al., 2021*). Primary antibodies (diluted 1:100) included anti-acetyl-histone H3K9 antibody (Abcam, ab12179), anti-di-methyl-histone H3K9 antibody (Abcam, ab1220), and anti-tri-methyl histone H3K4 (Millipore, 07-473). Secondary antibodies (diluted 1:1000) included goat anti-mouse IgG (H+L) Alexa Fluor 594 (Thermo

Fisher Scientific, A11005), goat anti-mouse IgG (H+L) Alexa Fluor 488 (Thermo Fisher Scientific, A11001), and goat anti-rabbit IgG (H+L) Alexa Fluor 568 (Thermo Fisher Scientific, A11011). Epifluorescence and DIC microscopy were performed using an Axio Imager M2 Microscope (Zeiss). Images were captured with an ORCA-ER digital camera (Hamamatsu) and processed using Axiovision software (Zeiss).

## Affinity chromatography of histidine-tagged SUMO

Synchronous adult worms (~200,000) were used for Ni-NTA pull-downs. HIS-tagged SUMO/SUMOylated proteins were enriched as described in *Kim et al., 2021*.

## Co-IP and western blotting

IPs were performed as described (*Kim et al., 2021*). For western blots, protein samples were denatured in NuPAGE LDS sample buffer (4×) (Thermo Fisher Scientific, NP0008), loaded on precast NuPAGE Novex 4–12% Bis-Tris protein gel (Life Technologies, NP0321BOX), and transferred onto 0.2 μm nitrocellulose membrane (Bio-Rad, 1704158) using a Transblot turbo transfer system (Bio-Rad). Membranes were incubated with primary antibodies at 4°C overnight and then with secondary antibodies for 1.5 hr at room temperature. Primary antibodies (and dilutions) included anti-FLAG (1:1000) (Sigma-Aldrich, F1804), anti-GFP (1:1000) (GenScript, A01704) anti-MRG-1 (1:1000) (Novus Biologicals, 49130002), anti-HPL-1 (1:1000) (Novus Biologicals, 38620002), anti-HPL-2 (1:1000) (Novus Biologicals, 38630002), anti-LIN-53 (1:1000) (Novus Biologicals, 48710002), anti-HDA-1 (1:2500) (Novus Biologicals, 38660002), anti-LET-418 (1:1000) (Novus Biologicals, 48960002), anti-SMO-1 (1:500) (purified from Hybridoma cell cultures, the Hybridoma cell line was gift from Ronald T. Hay, University of Dundee) (*Pelisch et al., 2014*), anti-tubulin (1:2000) (Cell Signaling Technology, 9099S), anti-ubiquitin (1:1000) (Abcam, ab7780), and anti-WAGO-9/HRDE-1 (1:500) (gift from Eric A. Miska) (*Ashe et al., 2012*). Antibody binding was detected with HRP-conjugated secondary antibodies: goat anti-mouse (1:2500) (Thermo Fisher Scientific, 62-6520) and mouse anti-rabbit (1:3000) (Abcam, ab99697).

## Immunoprecipitation-mass spectrometry (IP-MS)

Synchronous adult worms were used for IP experiments unless otherwise indicated. Worm lysates were prepared by combining 1 ml frozen worm pellet mixed with 1 ml IP lysis buffer (*Kim et al., 2021*) and 2 ml glass beads, and then vortexing on FastPrep (MP Biomedicals) at a speed of 6 m/s for 20 s for four times. Lysates were clarified twice by centrifugation for 30 min each at 12,000 rpm, 4°C. GBP beads were prepared by conjugating GBP nanobody to N-hydroxysuccinimide (NHS)-activated Sepharose beads (*Rothbauer et al., 2008*). GBP beads were incubated with lysates for 1 hr at 4°C on a rotating shaker, and GFP-tagged proteins and complexes were eluted with 1% sodium dodecyl sulfate (SDS), 50 mM Tris, pH 8.0 at 95°C for 10 min. Proteins were precipitated with trichloroacetic acid (TCA) and digested with trypsin. The resulting peptides were analyzed on a Q Exactive mass spectrometer (Thermo Fisher Scientific) coupled with an Easy-nLC1000 liquid chromatography system (Thermo Fisher Scientific). Peptides were loaded on a pre-column (75 μm ID, 6 cm long, packed with ODS-AQ 120 Å−10 μm beads from YMC Co., Ltd.) and separated on an analytical column (75 μm ID, 13 cm long, packed with Luna C18 1.8 μm 100 Å resin from Welch Materials) using a 78 min acetonitrile gradient from 0% to 30% (v/v) at a flow rate of 200 nl/min. The top 15-most intense precursor ions from each full scan (resolution 70,000) were isolated for HCD MS2 (resolution 17,500; NCE 27), with a dynamic exclusion time of 30 s. We excluded the precursors with unassigned charge states or charge states of 1+, 7+, or >7+ . Database searching was done by pFind 3.1 (http://pfind.ict.ac.cn/) against the *C. elegans* protein database (UniProt WS235). The filtering criteria were: 1% false discovery rate (FDR) at both the peptide level and the protein level; precursor mass tolerance, 20 ppm; fragment mass tolerance, 20 ppm; the peptide length, 6–100 amino acids. Spectral counts of HDA-1 in IP samples were used for normalization. Proteins either absent in N2 or with more than twofold of the spectra counts in the *hda-1::gfp* IP compared to those in N2 are shown in *Supplementary file 2*.

## Silver staining

One-third of each sample from each IP-MS experiment was fractionated on a 4–12% SDS-PAGE gel. Gels were silver stained using ProteoSilver Plus Silver Stain Kit (Sigma-Aldrich). Visible bands were cut for trypsin digestion and MS identification, and the most abundant protein in each band is labeled in *Figure 3A*.

## RNAi inheritance

L1 *oma-1::gfp* larvae were placed on IPTG plates (NGM plate with 1 mM IPTG and 100 μg/ml ampicillin) seeded with *E. coli* strain HT115 transformed with either control vector L4440 or *gfp* RNAi plasmid. The worms were scored for OMA-1::GFP signal in the oocyte at gravid adult stage and transferred to regular NGM plates. OMA-1::GFP was monitored at gravid adult stage for every generation until most worms recovered the expression of *oma-1::gfp*.

## Germline mortal assay

For each strain, 18 worms at L4 stage were singled and grown at 25℃. For each generation, the mothers were transferred to new plates every two days and their brood sizes were scored. One worm from each plate for every generation was randomly picked to continue scoring the brood size until plates became totally sterile.

## Gonad mRNA-seq and analysis

For every mutant, about 100 gonads were dissected from the young adult worms (first day as adults). We carefully cut near the −1 oocyte to make sure every gonad is similar and transfer gonads into a 1.5 ml microcentrifuge tube. RNA was extracted from dissected gonads using Tri-reagent with a yield about 0.5 μg. Ribosomal RNA was depleted by RNase H digestion after being annealed with home-made anti-rRNA oligos for *C. elegans*. DNA was removed by DNase treatment. RNA-seq libraries were constructed using a KAPA RNA HyperPrep kit, and paired-end sequencing was performed on a Nextseq 500 Sequencer with the illumina NextSeq 500/550 high-output kit v2.5 (150 cycles).

Salmon was used to map the mRNA-seq reads with the worm database WS268, and its output files were imported to DESeq2 in R. The differentially expressed genes were filtered by fold change more than 2 and adjusted p-value <0.05. The scatter plots were generated by the plot function in R.

## CHIP-seq

CHIP was carried out using a previous described protocol (*Askjaer et al., 2014*). Young adult worms were washed three times with M9 and once with PBS and then cross-linked with 1.1% formaldehyde in phosphate-buffered saline (PBS) with protease inhibitors for 10 min before being quenched with 125 mM glycine. Exchanging to CHIP Cell lysis buffer (20 mM Tris-Cl, 85 mM KCl, 0.5% NP40, pH 8.0), DNA was fragmented by sonication (Bioruptor, high intensity, 30 s on and 30 s off for 45 cycles). Samples were incubated with antibodies overnight at 4℃, and then with magnetic beads, which were precleaned with 5% bovine serum albumin (BSA) in PBS + 0.02% Tween-20 (PBST) for 4 hr. After a series of washes in buffers of different stringency, DNA was eluted with CHIP elution buffer (1% SDS, 250 mM NaCl, 10 mM Tris pH 8.0, 1 mM EDTA) at 65℃ for 15 min twice. RNA and proteins were removed by RNase A and Proteinase K treatments. Samples were reverse cross-linked by incubating at 65℃, and the DNA was purified using a Zymo DNA clean kit (cat # D5205). Libraries were prepared with NEBNext Ultra II DNA Library Prep Kit. Libraries were multiplexed and sequenced on HiSeq X or NovaSeq 6000 with paired-end 150 bp sequencing. With a pair-wised IP sample and its input, a ChIP-Seq Pipeline in the DolphinNext platform (built by the Bioinformatics core at UMass Medical School) was used to analyze the CHIP-seq data. The data processing and analysis pipeline includes adapter removal (cutadapt), reads mapping (Bowtie2-align-s v2.2.3), duplicates removal (Picard-tools v1.131), peak calling (MACS2 v2.1.1.20160309), peak location, and quantification (Bedtools v2.25.0, *Quinlan and Hall, 2010*). The output bed files were used to generate figures with IGV (v2.7.2) for *Figure 4D*. Background subtraction was applied with the intersect function of Bedtools for *Figure 5—figure supplement 3*.

## Small RNA cloning and data analysis

The small RNA cloning was conducted as the previous report (*Shen et al., 2018*). Worms were synchronized and collected at young adult stage. Small RNAs were enriched using a mirVana miRNA isolation kit (Invitrogen) from Trizol purified total RNA. Homemade PIR-1 was used to remove the d or triphosphate at the 5′ to generate 5′ monophosphorylated small RNA. Products were then ligated to 3′ adaptor (/5rApp/TGGAATTCTCGGGTGCCAAGG/3ddC/) by T4 RNA ligase 2(NEB) and 5′ adaptor (rGrUrUrCrArGrArGrUrUrCrUrArCrArGrUrCrCrGrArCrGrArUrCrNrNrNrCrGrArNrNrNrNrUrA rCrNrNrN, N is for a random nucleotide) by T4 ligase 1(NEB) sequentially, followed by reverse transcription with RT primer (CCTTGGCACCCGAGAATTCCA) and SuperScript III (Invitrogen). PCR amplification was done with Q5 and primers with indexes (forward: AATGATACGGCGACCACC-GAGATCTACACGTTCAGAGTTCTACAGTCCGA, reverse: CAAGCAGAAGACGGCATACGAGAT [6bases index] GTGACTGGAGTTCCTTGGCACCCGAGAATTCCA). PCR productions around 150 bp were separated by 12% SDS-PAGE, extracted with TE buffer (10 mM Tris-HCl, 0.1 mM EDTA, pH 8.0), and purified with isopropanol precipitation. Libraries were equally mixed and sequenced on a NextSeq 550 sequencer using the Illumina NextSeq 500/550 high-output kit v2.5 (75 cycles) with 75 bp single-end sequencing. Adapters were trimmed by cutadapt and reads were mapped to a worm database (WS268) using Bowtie2. DESeq2 was used to normalized reads between samples.

## Acknowledgements

We thank members of the Mello and Ambros labs for discussions. We thank M Shirayama for sharing strains, and the Miska and Hay labs for sharing antibodies. Onur Yukselen and Alper Kucukural from Bioinformatics Core at UMass Medical School provided support on ChIP-seq data analysis. CCM is a Howard Hughes Medical Institute Investigator. This work was supported in part by NIH Grant GM 58800.

## Additional information

### Funding

| Funder | Grant reference number | Author |
| --- | --- | --- |
| NIH Office of the Director | GM58800 | Craig C Mello |
| Howard Hughes Medical Institute | | Craig C Mello |

The funders had no role in study design, data collection and interpretation, or the decision to submit the work for publication.

### Author contributions

Heesun Kim, Yue-He Ding, Conceptualization, Data curation, Formal analysis, Validation, Investigation, Visualization, Methodology, Writing - original draft, Writing - review and editing; Gangming Zhang, Yong-Hong Yan, Investigation; Darryl Conte Jr, Writing - review and editing; Meng-Qiu Dong, Supervision, Investigation; Craig C Mello, Conceptualization, Supervision, Funding acquisition, Investigation, Writing - original draft, Writing - review and editing

### Author ORCIDs

Heesun Kim (iD) https://orcid.org/0000-0001-7643-4516
Darryl Conte Jr (iD) https://orcid.org/0000-0002-1137-8901
Meng-Qiu Dong (iD) http://orcid.org/0000-0002-6094-1182
Craig C Mello (iD) https://orcid.org/0000-0001-9176-6551

### Decision letter and Author response

Decision letter https://doi.org/10.7554/eLife.63299.sa1
Author response https://doi.org/10.7554/eLife.63299.sa2

# Additional files

## Supplementary files

- Supplementary file 1. Summary of RNAi-based genetic screen of chromatin factors and modifiers using a silenced piRNA sensor.
- Supplementary file 2. HDA-1 interactors identified from HDA-1::GFP immunoprecipitation-mass spectrometry (IP-MS).
- Supplementary file 3. List of *C. elegans* strains used in this study.
- Supplementary file 4. List of gRNA and ssOligo donor sequences.
- Supplementary file 5. RNA-seq data (deposited to Bioproject: PRJNA657279).
- Transparent reporting form

## Data availability

RNA Sequencing Data have been deposited in SRA under accession codes Bioproject: PRJNA657279. ChIP Sequencing Data have been deposited in SRA under accession codes Bioproject: PRJNA657194. All data generated or analyzed during this study are included in the manuscript and supplementary files.

The following datasets were generated:

| Author(s) | Year | Dataset title | Dataset URL | Database and Identifier |
| --- | --- | --- | --- | --- |
| Mello CC | 2021 | RNA seq of NuRD complex mutants and piRNA pathway mutants | https://www.ncbi.nlm.nih.gov/bioproject/PRJNA657279 | NCBI BioProject, PRJNA657279 |
| Mello CC | 2021 | ChIP seq of NuRD complex components and histone modifications | https://www.ncbi.nlm.nih.gov/bioproject/?term=PRJNA657194 | NCBI BioProject, PRJNA657194 |

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

# Appendix 1

**Appendix 1—key resources table**

| Reagent type (species) or resource | Designation | Source or reference | Identifiers | Additional information |
|---|---|---|---|---|
| Antibody | Mouse monoclonal anti-FLAG M2 | Sigma-Aldrich | Cat# F1804; RRID:AB_262044 | IB(1:1000) |
| Antibody | Rabbit polyclonal anti-GFP | GenScript | Cat# A01704; RRID:AB_2622199 | IB(1:1000) |
| Antibody | Rabbit polyclonal anti-MRG-1 | Novus Biologicals | Cat# 49130002; RRID:AB_10011724 | IB(1:1000) |
| Antibody | Rabbit polyclonal anti-HPL-1 | Novus Biologicals | Cat# 38620002; RRID:AB_10008562 | IB(1:1000) |
| Antibody | Rabbit polyclonal anti-HPL-2 | Novus Biologicals | Cat# 38630002; RRID:AB_10647696 | IB(1:1000) |
| Antibody | Rabbit polyclonal anti-LIN-53 | Novus Biologicals | Cat# 48710002; RRID:AB_10011629 | IB(1:1000) |
| Antibody | Rabbit polyclonal anti-HDA-1 | Novus Biologicals | Cat# 38660002; RRID:AB_10708816 | IB(1:2500) |
| Antibody | Rabbit monoclonal anti-tubulin | Cell Signaling Technology | Cat# 9099, RRID:AB_10695471 | IB(1:2000) |
| Antibody | Rabbit polyclonal anti-LET-418 | Novus Biologicals | Cat# 48960002, RRID:AB_10708820 | IB(1:1000) |
| Antibody | Mouse monoclonal anti-SMO-1 | *Pelisch et al., 2014* | Gift from Ronald T Hay | IB(1:500) freshly purified from hybridoma cell culture |
| Antibody | Rabbit polyclonal anti-ubiquitin | Abcam | Cat# ab7780; RRID:AB_306069 | IB(1:1000) |
| Antibody | Rabbit polyclonal anti-HRDE-1/WAGO-9 | *Ashe et al., 2012* | Gift from Eric A Miska | IB(1:500) |
| Antibody | Goat anti-mouse IgG (HRP-conjugated) | Thermo Fisher Scientific | Cat# 62-6520; RRID:AB_2533947 | IB(1:2500) |
| Antibody | Mouse Anti-rabbit IgG light (HRP-conjugated) | Abcam | Cat# ab99697; RRID:AB_10673897 | IB(1:3000) |
| Antibody | Mouse monoclonal anti-histone H3, di methyl K9 | Abcam | Cat# ab1220; RRID:AB_449854 | IF(1:100) |
| Antibody | Mouse monoclonal anti-histone H3, acetyl K9 | Abcam | Cat# ab12179; RRID:AB_298910 | IF(1:100) |
| Antibody | Rabbit polyclonal anti-trimethyl histone H3, K4 | Millipore | Cat# 07-473; RRID:AB_1977252 | IF(1:100) |

*Continued on next page*

*Appendix 1—key resources table continued*

| Reagent type (species) or resource | Designation | Source or reference | Identifiers | Additional information |
|---|---|---|---|---|
| Antibody | Goat anti-mouse IgG (H+L) Alexa Fluor 488 | Thermo Fisher Scientific | Cat# A-11001; RRID:AB_2534069 | IF(1:1000) |
| Antibody | Goat anti-mouse IgG (H+L) Alexa Fluor 594 | Thermo Fisher Scientific | Cat# A-11005; RRID:AB_2534073 | IF(1:1000) |
| Antibody | Goat anti-rabbit IgG (H+L) Alexa Fluor 568 | Thermo Fisher Scientific | Cat# A-11011; RRID:AB_143157 | IF(1:1000) |
| Antibody | Mouse monoclonal anti-FLAG | Sigma-Aldrich | Cat# F3165; RRID:AB_259529 | For CHIP |
| Antibody | Mouse monoclonal anti-dimethyl histone H3, K9 | MBL international | Cat# MABI0307; RRID:AB_11124951 | For CHIP |
| Antibody | Rabbit polyclonal anti-histone H3, trimethyl K9 | Millipore | Cat# 07-523; RRID:AB_310687 | For CHIP |
| Antibody | Rabbit polyclonal anti-histone H3, acetyl K9 | Abcam | Cat# ab4441; RRID:AB_2118292 | For CHIP |
| Strain, strain background | *C. elegans* strains | This study | | *Supplementary file 3* |
| Strain, strain background | *E. coli*: Strain OP50 | Caenorhabditis Genetics Center | WormBase: OP50 | |
| Strain, strain background | *E. coli*: Strain HT115 | *Caenorhabditis* Genetics Center | WormBase: HT115 | |
| Strain, strain background | *E. coli*: Ahringer collection | Laboratory of C. Mello | N/A | |
| Peptide, recombinant protein | Ex Taq DNA polymerase | Takara | Cat# RR001C | |
| Peptide, recombinant protein | iProof high fidelity DNA polymerase | Bio-Rad | Cat#1725302 | |
| Peptide, recombinant protein | Alt-R S.p. Cas9 Nuclease V3 | Integrated DNA Technologies (IDT) | Cat# 1081058 | CRISPR reagent |
| Peptide, recombinant protein | Alt-R A.s. Cas12a (Cpf1) V3 | Integrated DNA Technologies (IDT) | Cat# 1081068 | CRISPR reagent |
| Peptide, recombinant protein | GFP-binding protein (GBP) beads | Homemade | N/A | |
| Peptide, recombinant protein | Hybridase Thermostable RNase H | Lucigen Corporation | Cat# H39500 | |
| Peptide, recombinant protein | Turbo DNase | Thermo Fisher Scientific | Cat# AM2238 | |
| Peptide, recombinant protein | Super Script III Reverse Transcriptase | Thermo Fisher Scientific | Cat# 18080085 | |

*Continued on next page*

*Appendix 1—key resources table continued*

| Reagent type (species) or resource | Designation | Source or reference | Identifiers | Additional information |
|---|---|---|---|---|
| Peptide, recombinant protein | T4 RNA ligase 1 | New England Biolabs | Cat# M0437M | |
| Peptide, recombinant protein | T4 RNA ligase 2 | New England Biolabs | Cat# M0242L | |
| Peptide, recombinant protein | Super Script III Reverse Transcriptase | Thermo Fisher Scientific | Cat# 18080085 | |
| Chemical compound, drug | Ethidium bromide | Sigma-Aldrich | Cat# E1510 | |
| Chemical compound, drug | Isopropyl-$\beta$ -D-thiogalactoside (IPTG) | Sigma-Aldrich | Cat# 11411446001 | |
| Chemical compound, drug | Ampicillin | Sigma-Aldrich | Cat# A9518 | |
| Chemical compound, drug | Tetracycline | Sigma-Aldrich | Cat# 87128 | |
| Chemical compound, drug | Indole-3-acetic acid (IAA) | Alfa Aesar | Cat# A10556 | |
| Chemical compound, drug | Tetramisole hydrochloride | Sigma-Aldrich | Cat# L9756-5G | |
| Chemical compound, drug | Paraformaldehyde 16% solution | Electron Microscopy Science | Cat# Nm15710 | |
| Chemical compound, drug | Formaldehyde, 36.5–38% in $H_2O$ | Sigma-Aldrich | Cat# F8775 | |
| Chemical compound, drug | PBS | Life Technologies | Cat# AM9615 | |
| Chemical compound, drug | Tween20 | Fisher BioReagents | Cat# BP337-500 | |
| Chemical compound, drug | Bovine serum albumin (BSA) | Life Technologies | Cat# AM2618 | |
| Chemical compound, drug | 1M HEPES, pH7.4 | TEKnova | Cat# H1030 | |
| Chemical compound, drug | Sodium citrate dihydrate | Thermo Fisher Scientific | Cat# BP337500 | |
| Chemical compound, drug | Triton X-100 | Sigma-Aldrich | Cat# T8787-250ml | |
| Chemical compound, drug | cOmplete EDTA-free Protease Inhibitor Cocktail | Roche | Cat# 11836170001 | |
| Chemical compound, drug | NP-40 | EMD Millipore | Cat# 492018 | |
| Chemical compound, drug | Tris (Base) | Avantor | Cat# 4099–06 | |
| Chemical compound, drug | Ethylenediaminetetraacetic acid disodium salt dihydrate | Sigma-Aldrich | Cat# E1644 | |
| Chemical compound, drug | TE buffer, pH 8.0 | Thermo Fisher Scientific | Cat# AM9858 | |
| Chemical compound, drug | Sodium dodecyl sulfate (SDS) | Sigma-Aldrich | Cat# L3771-100G | |
| Chemical compound, drug | Sodium chloride (NaCl) | Genesee Scientific | Cat# 18-214 | |

*Continued on next page*

*Appendix 1—key resources table continued*

| Reagent type (species) or resource | Designation | Source or reference | Identifiers | Additional information |
|---|---|---|---|---|
| Chemical compound, drug | Magnesium chloride (MgCl$_2$) | Sigma-Aldrich | Cat# M8266 | |
| Chemical compound, drug | DL-Dithiothreitol (DTT) | Sigma-Aldrich | Cat# D0632-10G | |
| Chemical compound, drug | Calcium chloride (CaCl$_2$) | Sigma-Aldrich | Cat# C5080 | |
| Chemical compound, drug | Potassium chloride (KCl) | Sigma-Aldrich | Cat# P9541 | |
| Chemical compound, drug | Guanidine-HCl | Sigma-Aldrich | Cat# G3272 | |
| Chemical compound, drug | Imidazole | Sigma-Aldrich | Cat# 792527 | |
| Chemical compound, drug | $\beta$-Mercaptoethanol | Sigma-Aldrich | Cat# M6250 | |
| Chemical compound, drug | Sodium phosphate, dibasic | Sigma-Aldrich | Cat# S7907 | |
| Chemical compound, drug | Sodium phosphate, monobasic | Sigma-Aldrich | Cat# S0751 | |
| Chemical compound, drug | Urea | Thermo Fisher Scientific | Cat# Ac327380010 | |
| Chemical compound, drug | Trichloroacetic acid (TCA) | Sigma-Aldrich | Cat# T0699 | |
| Chemical compound, drug | 1-Bromo-3-chloropropane | Sigma-Aldrich | Cat# B9673 | |
| Chemical compound, drug | Glycine | Thermo Fisher Scientific | Cat# BP381-1 | |
| Chemical compound, drug | TRI reagent | Sigma-Aldrich | Cat# T9424 | |
| Chemical compound, drug | Trypsin | New England Biolabs | Cat# P8101S | |
| Commercial assay, kit | SlowFade Diamond antifade Mountant with DAPI | Life Technologies | Cat# S36964 | |
| Commercial assay, kit | Quick start Bradford 1xdye reagent | Bio-Rad | Cat# 5000205 | |
| Commercial assay, kit | NuPAGE LDS sample buffer (4x) | Thermo Fisher Scientific | Cat# NP0008 | |
| Commercial assay, kit | GlycoBlue Coprecipitant | Thermo Fisher Scientific | Cat# AM9515 | |
| Commercial assay, kit | Ni-NTA resin | Qiagen | Cat# 30210 | |
| Commercial assay, kit | pCR-Blunt II Topo cloning kit | Thermo Fisher Scientific | Cat# K280020 | |
| Commercial assay, kit | MinElute PCR purification Kit | Qiagen | Cat# 28006 | |
| Commercial assay, kit | ChIP DNA clean and concentrator Kit | Zymo Research | Cat# 5205 | |
| Commercial assay, kit | ProteoSilver Plus Silver Stain Kit | Sigma-Aldrich | Cat# PROT-SIL2 | |
| Commercial assay, kit | Trans-blot Turbo Mini NC Transfer Packs | Bio-Rad | Cat# 1704158 | |

*Continued on next page*

*Appendix 1—key resources table continued*

| Reagent type (species) or resource | Designation | Source or reference | Identifiers | Additional information |
|---|---|---|---|---|
| Commercial assay, kit | Lumi-Light Plus western blotting substrate | Sigma-Aldrich | Cat# 12015196001 | |
| Commercial assay, kit | Hyperfilm ECL | Thermo Fisher Scientific | Cat# 45001507 | |
| Commercial assay, kit | *mir*Vana miRNA Isolation Kit | Thermo Fisher Scientific | Cat# AM1561 | |
| Commercial assay, kit | KAPA RNA HyperPrep with RiboErase (KK8560) | Roche | Cat# 08098131702 | |
| Commercial assay, kit | KAPA single-indexed adapter kit (KK8700) | Roche | Cat# 08005699001 | |
| Commercial assay, kit | ChIP-Grade Protein A/G Magnetic Beads | Thermo Fisher Scientific | Cat# 26162 | |
| Commercial assay, kit | Illumina NextSeq 500/550 v2.5 kit (75 cycles) | Illumina, Inc. | Cat# 20024906 | |
| Commercial assay, kit | Illumina NextSeq 500/550 v2.5 kit (150 cycles) | Illumina, Inc. | Cat# 20024907 | |
| Recombinant DNA reagent | Peft3::cas9 vector (backbone: blunt II topo vector) | *Friedland et al., 2013* | N/A | Backbone: blunt II topo vector (*Kim et al., 2021*) |
| Recombinant DNA reagent | pRF4: injection marker, *rol-6 (su1006)* | *Mello et al., 1991* | N/A | Backbone: blunt II topo vector (*Kim et al., 2021*) |
| Recombinant DNA reagent | *smo-1* sgRNA plasmid | This study | | *Supplementary file 4* |
| Recombinant DNA reagent | mep-1 sgRNA plasmid | *Kim et al., 2021* | | *Supplementary file 4* |
| Sequence-based reagent | List of gRNA sequences | This study | | *Supplementary file 4* |
| Sequence-based reagent | Alt-R CRISPR-Cas9 tracrRNA | Integrated DNA Technologies (IDT) | Cat# 1072534 | CRISPR reagent |
| Sequence-based reagent | Anti-rRNA Oligos for *C. elegans* | This study (homemade) | N/A | |
| Software, algorithm | GraphPad Prism version 8.2.1 | GraphPad Software | http://www.graphpad.com | |
| Software, algorithm | Salmon | *Patro et al., 2017* | v1.1.0 | |
| Software, algorithm | DEseq2 | *Love et al., 2014* | v1.26.0 | |
| Software, algorithm | Bowtie2 | *Langmead and Salzberg, 2012* | v2.2.3 | |
| Software, algorithm | Picard-tools | *Broad Institute, 2019* | v1.131 | |
| Software, algorithm | MACS2 | *Feng et al., 2012* | v2.1.1.20160309 | |
| Software, algorithm | BedTools | *Quinlan and Hall, 2010* | v2.25.0 | |
| Software, algorithm | IGV | *Robinson et al., 2017* | v2.7.2 | |

