## [Decision Letter]

**Acceptance summary:**

These studies reveal that sumoylation of HDA-1, a type 1 histone deacetylase (HDAC), plays a key role in establishing transcriptional silencing of piRNA-regulated genes in *C. elegans*. Genetic analysis provides strong evidence that the SUMO pathway is involved in piRNA silencing, and further mechanistic analysis demonstrates this involves sumoylation of two lysines in the tail of HDA-1. HDA-1 sumoylation promotes its association with the NuRD chromatin modifier complex, which enhances local H3K9ac deacetylation, resulting in negative regulation of hundreds of target genes that is instrumental in the inherited RNAi pathway.

**Decision letter after peer review:**

Thank you for submitting your article "HDAC1 SUMOylation promotes Argonaute directed transcriptional silencing in *C. elegans*" for consideration by *eLife*. Your article has been reviewed by 3 peer reviewers, including Tony Hunter as the Reviewing Editor and Reviewer #1, and the evaluation has been overseen by Kevin Struhl as the Senior Editor. The following individual involved in review of your submission has agreed to reveal their identity: Federico Pelisch (Reviewer #2).

The reviewers have discussed the reviews with one another and the Reviewing Editor has drafted this decision to help you prepare a revised submission.

Summary:

In this paper, your studies showed that sumoylation of HDA-1, a type 1 HDAC, at two C-terminal Lys residues plays a role in establishing transcriptional silencing of piRNA-regulated genes in *C. elegans* through enhanced NuRD complex interaction and histone H3 deacetylation. The reviewers all found the link between HDA-1 sumoylation and silencing to be interesting, but raised a number of issues that need to be addressed. In particular, more mechanistic insights into

Essential revisions:

1. The evidence that sumoylation of HDA-1 increases its silencing function is reasonably strong, but no direct evidence is provided that HDA-1 sumoylation at K444 and K459 increases its deacetylase activity, as is proposed. For this purpose, biochemical assays comparing unmodified HDA-1 (desumoylated WT or KKRR mutant HDA-1) with sumoylated HDA-1 (or SUMO-fused HDA-1) to measure changes in deacetylatase activity are required. In this connection, there are also no mechanistic insights into how the presence of a SUMO moiety at either K444 or K459 site in the C-terminal tail would increase HDA-1 catalytic activity. Finally, your evidence for sumoylation is based on K to R mutations of these two Lys, and direct mass spec evidence that the K444 and K459 sites are in fact sumoylated in vivo would be desirable, although not essential.

2. Although the evidence that sumoylated HDA-1 interacts with the NuRD complex is reasonable, it remains unclear how sumoylation of HDA-1 increases its recognition by the NuRD complex. To determine whether the HDA-1 SUMO moieties are directly involved in the interaction, testing for SIMs (and/or UIMs) in MEP-1, or another subunit of the NuRD complex subunit is needed, followed by verification of the importance of these residues in vivo or through in vitro interaction assays.

3. Evidence is provided that HDA-1 sumoylation decreases total H3K9ac histone acetylation in vivo, but the use of ChIP-seq to test if HDA-1 and NuRD complex recruitment, and local histone H3 acetylation are affected at known target genes in the KKRR mutant and HDA-1 degron strains versus the wild type strain, with correlated RNA-seq data in the same lines, would significantly strengthen the conclusion that HDA-1 sumoylation is required for piRNA regulation. The addition of ChIP-seq data for HDA-1 and HDA-1-KKRR genome-wide distribution profiles would further strengthen the authors' claims, but is not essential.

4. A possible role of the GEI-17 E3 SUMO ligase in the regulation of HDA-1 and piRNA-regulated gene silencing was excluded through the use of a gei-17 mutant. However, while neither gei-17 nor pie-1 Individually affected piRNA reporter expression in P0, the authors need to test a double mutant to confirm a lack of synergy.

5. Based on the data presented only a small fraction of the HDA-1 population was sumoylated, and the authors need discuss how sumoylation of such a small fraction of the HDA-1 protein population would be able to exert the functional effects on HDA-1 activity they observed.

6. The authors could do a more comprehensive job of reviewing the relevant literature on the role of sumoylation in Piwi-mediated silencing in other organisms, e.g. Ninova et al. Mol Cell 77:556, 2020.

Additional points are raised in the individual reviews below and could be considered, if the authors believe that addressing them would strengthen the manuscript.

*Reviewer #1:*

The evidence that sumoylation of HDA-1, a type 1 HDAC, plays a key role in establishing transcriptional silencing of piRNA-regulated genes in *C. elegans* is quite convincing. The genetic analysis demonstrating that the SUMO pathway is involved in piRNA silencing is strong, and the mutational evidence that this involves sumoylation of two Lys in the tail of HDA-1 is reasonable. Likewise, the finding that HDA-1 sumoylation promotes association with NuRD complex components and association of MEP-1, an HDA-1 interactor, with chromatin regulators is convincing. In addition, the evidence that HDA-1 sumoylation increases H3K9ac deacetylation in vivo, leading to negative regulation of hundreds of target genes, and plays a role in the inherited RNAi pathway is solid.

While the overall conclusion provides an interesting advance in understanding mechanisms of piRNA-mediated gene silencing in *C. elegans*, the paper is lacking any biochemical analysis of the effects of sumoylation on HDA-1 activity and its association with other transcriptional regulators.

1. The authors mapped two sumoylation sites close to the C terminus of HDA-1, K444 and K459, based on extremely weak homology with two established sumoylation sites in human HDAC1 that are reported to be important for transcriptional repression (N.B. the authors should indicate here that David et al. reported that K444/476R HDAC1 had reduced transcriptional repression activity in reporter assays.). While the two human sites conform to the sumoylation site consensus, ψKXE, neither K444 nor K459 in HDA-1 fits this consensus (possibly one could argue that K444 is in an inverted motif). The fact that the KKRR mutant HDA-1 is no longer sumoylated is consistent with these two Lys being sumoylated, but it would be reassuring to have direct MS evidence that K444 and K459 are indeed sumoylated, which could be achieved using a SUMO Thr91Arg mutant that generates a GlyGly stub upon trypsin digestion, among other methods.

2. It remains unclear how sumoylated HDA-1 is recognized by MEP-1 for assembly into the NuRD complex. Does MEP-1, or another NuRD subunit, have a SIM that could facilitate direct interaction of MEP-1 and sumoylated HDA-1?

3. As the authors discuss, it is surprising that the HDA-1(KKRR)::SUMO protein, which in effect is a constitutively sumoylated form of HDA-1 that will interact constitutively with MEP-1/NuRD, does not have more deleterious effects on the organism, since according to the data in Figure 2B, the stoichiometry of endogenous HDA-1 sumoylation was extremely low. Of course, low sumoylation stoichiometry, which is a general issue with sumoylation studies, means that only a very small fraction of the HDA-1 endogenous population will be able to engage with the silencing complexes at any one time. This point is also worth discussion.

4. Page 5: Here, and elsewhere, the authors claim that sumoylation of the two C-terminal Lys activates HDA-1 histone deacetylase activity, but provide no direct evidence for this statement. There are no HDAC assays, and it is unclear how C-terminal SUMO residues distant from the catalytic domain would alter its enzymatic activity, unless there is a SIM motif in HDA-1 that might allow for intramolecular interaction with SUMO residues at the tail leading to a conformation change. Did the authors check for a SIM motif in HDA-1? The fact that adding SUMO to the C-terminus rather than one or both of the two Lys would also have to be taken into account in determining bow sumoylation might "activate" HDA-1. To demonstrate that sumoylation activated HAD-1 in vitro deacetylation assays would need to be carried out comparing the activities of unmodified and sumoylated HDA-1. Instead of enzymatic activation, it is possible that the PIE-1 interaction and HDA-1 sumoylation results in relocalization of HDA-1 within the nucleus to facilitate more efficient H3K9ac deacetylation.

*Reviewer #2:*

In their manuscript, Kim et al. describe a role for HDAC1 (HDA-1) sumoylation in Argonaute-directed transcriptional silencing. The authors suggest that sumoylation of HDA-1 is important for proper assembly of the NuRD deacetylase complex. The role of SUMO modification in heterochromatin has been extensively documented and it is a very interesting topic. The current manuscript provides a very interesting set of results on this topic. I have list of comments, suggestions, questions, and concerns, which are listed below, especially related to the first half of the results:

1. A general question would be how can HDA-1 sumoylation, which is barely detectable, account for such a big 'positive' effect on complex assembly? HDA-1 SUMO modification seems around 10% after enriching for SUMO-modified proteins, which means that stoichiometry will be way lower than this. While this is common for SUMO-modified proteins, it does make it difficult to associate with a 'simple' model.

2. In Figure 1, a schematic of the sensor used throughout the study would benefit the reader.

3. In Figure 1, have the authors checked if the 10xHis::tagged smo-1 has the same effect as the 3xflag::smo-1 (i.e. is it also a partial loss of function allele)?

4. In Figure 1 it would be nice to see the global SUMO conjugation levels in the different conditions, particularly in the smo-1(RNAi), 3xflag::smo-1, and ubc-9(G56R).

5. Also Figure 1, was gei-17 depletion/deletion checked in any way (i.e. WB)? Did the authors consider other SUMO E3 ligase, such as the mms-21 orthologue?

6. While I am not a big fan of fusing SUMO to proteins, in this case it seems like a very reasonable thing to do, considering the modification sites are located very close to the C-terminal end of the protein. Did the authors check an N-terminal fusion?

7. In Figure 2B, it becomes very clear that the level of SUMO modification of HDA-1 is extremely small, barely detectable after an enrichment method. I also wonder why the gels were cropped so tightly, especially considering that in Figure 3 there is an additional band corresponding to ubiquitylated, sumoylated HDA-1. in vitro modification assays would be helpful. HDA-1 alongside a known and characterised SUMO substrate would indicate how good a substrate HDA-1 is.

8. In Figure 2D, is the difference between HDA-1(KKRR)::SUMO and HDA-1::SUMO significant?

9. In Figure 3A-C, it would be useful to control whether the GFP::HDA-1 fusion behaves as the untagged one in the sensor assay (wt vs. KKRR).

10. I have a few questions regarding Figure 3D:

i. Considering the extremely low level of HDA-1 sumoylation, did the authors detect SUMO and ub conjugated HAD-1 (not the SUMO usion)?

ii. Is ub conjugated to SUMO or to HDA-1?

iii. Does MEP-1 contain any obvious SIMs and or UIMs?

iv. To make a stronger case for the SUMO-dependent interaction model, in vitro interaction assays with recombinant proteins would be extremely useful.

11. In the discussion, the authors compare the lack of requirement for GEI-17 in their manuscript with the requirement for Su(var)2-10 in flies. It is very important to back this claim that the authors control GEI-17 depletion (as pointed out in 5).

Overall, I think this manuscript provides a very interesting set of results and I believe that, with the addition of some simple biochemical experiments, the quality and impact of the overall work would be much greater.

*Reviewer #3:*

This manuscript by Kim et al. describes a role of SUMOylation in Argonaute-directed transcriptional silencing in *C. elegans*. The Authors found that SUMOylation of the histone deacetylase HDA-1 promotes its interaction with both the Argonatue target recognition complex as well as the chromatin remodeling NuRD complex. This enables initiation of target silencing. Impaired SUMOylation of HDA-1 leads to loss of interactions with several protein complexes, reduced silencing of piRNA targets, and reduced brood size. While the findings and claims are interesting, some of the novelty is overemphasized and some of the claims are not fully supported by the data.

1. The importance of HDA-1 SUMOylation for transcriptional repression. The title "HDAC1 SUMOylation promotes Argonaute directed transcriptional silencing in *C. elegans*" implies a central role of SUMOylation in piRNA-mediated transcriptional silencing. The Argonaute HRDE-1/WAGO-9 targets countless transposons as shown previously and also in this manuscript (Figure S3), and so do the HDA-1 degron and Ubc9 mutant, indicating that histone deacetylation and protein SUMOylation are essential processes in TE silencing. However, the HDA-1 SUMOylation mutant (KKRR) only slightly affects 6 TE families (Figure S3), indicating that SUMOylation of HDA-1 might not be a key mediator of this process. Furthermore, the authors write that "Our findings suggest how SUMOylation of HDAC1 promotes the recruitment and assembly of an Argonaute-guided chromatin remodeling complex to orchestrate de novo gene silencing in the *C. elegans* germline.", but then they also state that "Comparison with mRNA sequencing data from auxin-treated degron::hda-1 animals revealed an even more extensive overlap with Piwi pathway mutants (Figure S2B), indicating that HDA-1 also promotes target silencing independently of HDA-1 SUMOylation." Based on their results and their own interpretations, I find that the importance of HDA-1 SUMOylation in piRNA-dependent transcriptional silencing is overemphasized.

Additionally, the model (Figure 7) implies that for initiation of silencing WAGO recruits HDA-1 to targets. This should be tested by analyzing HDA-1 distribution over WAGO targets in WT and upon loss of WAGO.

2. The mechanistic role of HDA-1 SUMOylation.

On page 17 (amongst other places) the authors claim that "The SUMOylation of HDA-1 promotes its activity, while also promoting physical interactions with other components of a germline nucleosome-remodeling histone deacetylase (NuRD) complex, as well as the nuclear Argonaute HRDE-1/WAGO-9 and the heterochromatin protein HPL-2 (HP1)".

• Regarding activity: Loss of deacetylation/silencing in the SUMO mutant might be due to loss of enzymatic activity, but it might also be due to defects in recruitment/complex formation. There is no data that proves altered enzymatic activity. In fact, Figure 6 indicates SUMO-dependent interaction of WAGO-9 with HDA-1, implying that recruitment is affected. To distinguish between activity and recruitment, at the very least, the authors would need to show that HDA-1 localization to its genomic targets is unaltered upon mutating its SUMOylation site (ChIP-seq of wt and KKRR mutant), while H3K9ac is increased (K9ac ChIP-seq in wt and KKRR mutant) in the mutant. This, in combination with HDA-1 localization in wt and WAGO-9 loss would imply whether complex formation to recruit HDA-1 or HDA-1 enzymatic activity is mostly affected by SUMOylation.

• Regarding physical interactions: Figure 3D shows that if we fuse a SUMO residue to HDA-1, it will interact with MEP-1, while SUMOylation deficient HDA-1 mutant doesn't interact. However, for the WT HDA-1 control, we only see unSUMOylated protein interacting with MEP-1. Furthermore, in the MEP-1 IPs of samples that should contain SUMO-fused HDA-1, the authors detect a lot of "cleaved", unSUMOylated HDA-1. Unless cleavage happened after IP, during elution (unlikely, and there is "cleaved" HDA-1 in the inputs), these findings argue that the interaction with MEP-1 is not mediated by HDA-1 SUMOylation. An interaction between MEP-1 and unmodified HDA-1 is also shown in the accompanying manuscript, which appears to be dependent on Pie-1 SUMOylation. Thus, SUMOylation of HDA-1 alone seems unlikely to be the major factor necessary for silencing complex assembly. (as a side question: Does the protease inhibitor cocktail used inhibit de-SUMOylation enzymes? I am concerned that deSUMOylating enzymes might compromise some result interpretations.)

• Regarding functional relevance of HDA-1 acetylation: On pages 12/13 authors claim that because "HDA-1(KKRR) animals and mep-1-depleted worms revealed dramatically higher levels of H3K9Ac compared to wild-type" and "HDA-1, LET-418/Mi-2, and MEP-1 bind heterochromatic", "SUMOylation of HDA-1 appears to drive formation or maintenance of germline heterochromatin regions of the genome." These correlations do not prove function. The authors have performed H3K9me2 (although not H3K9-ac) ChIP-seq in WT, KKRR mutant and HDA-1 degron worms, yet do not analyze globally whether acetylation is lost on genes that are affected (change in RNA-seq vs. change in K9me2 or acetyl). To support the claim that SUMOylation of HDA-1 drives deacetylation and heterochromatin formation, it would be important to show changes in H3K9Ac levels (or other acetyl marks) and potentially NuRD component occupancy between control and HDA-1 SUMOylation-deficient animals at specific targets (i.e. genes derepressed upon loss of SUMOylation identified in RNA-seq, and the reporter locus).

3. The authors claim (p17) that "initiation of transcriptional silencing requires SUMOylation of conserved C-terminal lysine residues in the type-1 histone deacetylase HDA-1". I do not see any supporting data that has separately looked at formation/initiation and maintenance of silencing (a technically challenging experiment).

4. The authors repeatedly claim that gei-17 does not play a role in piRNA target silencing, based on loss of gei-17 not affecting the piRNA reporter (Figure 1B). At the same time, they claim that pie-1 plays a role, even though it likewise does not affect the piRNA reporter (it affects the reporter only in F3; data on gei-17 effect in F3 is not present). In the accompanying paper, the authors show that while gei-17 loss by itself causes only moderate effect on extra intestine cells, combined with Pie-1 loss the effect is more severe than when Pie-1 loss is combined with Ubc9 or smo loss. This to me indicates an important role of gei-17 in inhibiting differentiation of germline stem cells to somatic tissues, but these effects are likely synergistic and thus masked by Pie-1. Individually neither Gei-17 nor Pie-1 show an effect on piRNA reporter in P0, but to confirm lack of synergy, their effects of should be tested together. Although possible, the present data is insufficient to rule of gei-17 involvement.

---

## [Author Response]

Essential revisions:1. The evidence that sumoylation of HDA-1 increases its silencing function is reasonably strong, but no direct evidence is provided that HDA-1 sumoylation at K444 and K459 increases its deacetylase activity, as is proposed. For this purpose, biochemical assays comparing unmodified HDA-1 (desumoylated WT or KKRR mutant HDA-1) with sumoylated HDA-1 (or SUMO-fused HDA-1) to measure changes in deacetylatase activity are required. In this connection, there are also no mechanistic insights into how the presence of a SUMO moiety at either K444 or K459 site in the C-terminal tail would increase HDA-1 catalytic activity.

As noted in our reply to the PIE-1 paper, we did not mean to imply that SUMO modification of HDAC increases its enzymatic activity. This possibility had not even occurred to us and seems very unlikely. We now emphasize the genetic and molecular genetic studies as support for the importance of direct HDAC SUMOylation in Argonaute-mediated germline surveillance. We have opted not to attempt in vitro HDAC assays, as it is not clear to us what additional insights any unlikely changes in activity might have. Moreover, such a study would very likely require structural investigations of any changes to the protein active site. We have now made it clear throughout the paper that the SUMO modifications are thought to alter HDAC activity indirectly by regulating its association with co-factors.

Finally, your evidence for sumoylation is based on K to R mutations of these two Lys, and direct mass spec evidence that the K444 and K459 sites are in fact sumoylated in vivo would be desirable, although not essential.

Unfortunately, MS detection of SUMOylated sites requires a mutation in *smo-1* that is lethal in worms. We mention this negative result in the PIE-1 paper.

2. Although the evidence that sumoylated HDA-1 interacts with the NuRD complex is reasonable, it remains unclear how sumoylation of HDA-1 increases its recognition by the NuRD complex. To determine whether the HDA-1 SUMO moieties are directly involved in the interaction, testing for SIMs (and/or UIMs) in MEP-1, or another subunit of the NuRD complex subunit is needed, followed by verification of the importance of these residues in vivo or through in vitro interaction assays.

**Thank you for raising this interesting point regarding SIMs in MEP-1. Our results clearly show that HDA-1-SUMO promotes its interactions with the NuRD complex. Whether or not it does this through the SIMs in MEP-1 or another NuRD complex factor is not relevant to our model or story. We have generated and characterized double-SIM mutants of MEP-1. Interestingly, the MEP-1 double-SIM mutant shows a strong desilencing of the piRNA sensor, but the worms are sterile due to an oogenesis defect—like the *mep-1* null mutant. Interpreting this phenotype is complicated by the likelihood that the SIMs on MEP-1 interact with not only HDA-1 but also other SUMOylated proteins. Future studies will have to address this important detail.**

**The issue of how HDA-1 SUMoylation promotes its interactions in the adult germline are now directly discussed as follows:**

“Paradoxically in the embryo HDA-1 is not SUMOylated and yet interacts robustly with MEP-1.[…] Thus although the SIM motifs in MEP-1 may be important for its interaction with HDA-1-SUMO they may also be required for other functions or for interactions with other essential co-factors.”3. Evidence is provided that HDA-1 sumoylation decreases total H3K9ac histone acetylation in vivo, but the use of ChIP-seq to test if HDA-1 and NuRD complex recruitment, and local histone H3 acetylation are affected at known target genes in the KKRR mutant and HDA-1 degron strains versus the wild type strain, with correlated RNA-seq data in the same lines, would significantly strengthen the conclusion that HDA-1 sumoylation is required for piRNA regulation. The addition of ChIP-seq data for HDA-1 and HDA-1-KKRR genome-wide distribution profiles would further strengthen the authors' claims, but is not essential.

**The genetic evidence that HDA-1 SUMOylation promotes Argonaute-mediated silencing is quite strong, and is supported by our RNA-seq data on dissected gonads and by the heritable silencing defect of HDA-1(KKRR) worms in response to a dsRNA trigger. ChIP-seq data would be nice to have, but would require purifying germline nuclei away from the somatic tissues of adults—a technical challenge. A few dozen dissected gonads would yield insufficient material for ChIP-seq experiments.**

4. A possible role of the GEI-17 E3 SUMO ligase in the regulation of HDA-1 and piRNA-regulated gene silencing was excluded through the use of a gei-17 mutant. However, while neither gei-17 nor pie-1 Individually affected piRNA reporter expression in P0, the authors need to test a double mutant to confirm a lack of synergy.

**Thank you for this excellent suggestion. Indeed, the double shows a very strong de-silencing phenotype as indicated in Figure 6 of the PIE-1 paper. We had trouble making this double due to a haploinsufficiency, but finally succeeded using CRISPR.**

5. Based on the data presented only a small fraction of the HDA-1 population was sumoylated, and the authors need discuss how sumoylation of such a small fraction of the HDA-1 protein population would be able to exert the functional effects on HDA-1 activity they observed.

**We apologize for neglecting to discuss this important point. The second section of the discussion now addresses this issue at some length:**

**“**SUMO, a potent genetic modifier with an elusive biochemical signature**Given the strong genetic evidence that the modification of HDA-1 by SUMO promotes piRNA silencing in the adult germline, we were surprised that the conjugated isoform of HDA-1 was undetectable in our IP assays from adult animals. […] Consistent with this idea, an HDA-1::SMO-1 translational fusion protein with an abnormal linkage via the N-terminus of SUMO was stable in protein lysates, strongly rescued the silencing defects of the presumptive SUMO-acceptor mutant protein HDA-1(KKRR), and dramatically enhanced the detection of protein-protein interactions between HDA-1 and components of an adult-stage NuRD complex.”**6. The authors could do a more comprehensive job of reviewing the relevant literature on the role of sumoylation in Piwi-mediated silencing in other organisms, e.g. Ninova et al. Mol Cell 77:556, 2020.

**We agree. We now review the literature more comprehensively in the final section of the discussion as follows:**

**“**Parallels in the role of SUMOylation in Piwi silencing in insects, mice and worms**Histone deacetylation is a necessary step in de novo transcriptional silencing. Yet, precisely how nuclear Argonautes orchestrate both deacetylation and the subsequent installation of silencing marks on target chromatin is not known. […]It will be interesting in the future to learn if mammalian and insect Piwi Argonautes target HDAC1 SUMOylation to promote de novo Piwi silencing”.**Additional points are raised in the individual reviews below and could be considered, if the authors believe that addressing them would strengthen the manuscript.Reviewer #1:The evidence that sumoylation of HDA-1, a type 1 HDAC, plays a key role in establishing transcriptional silencing of piRNA-regulated genes in *C. elegans* is quite convincing. The genetic analysis demonstrating that the SUMO pathway is involved in piRNA silencing is strong, and the mutational evidence that this involves sumoylation of two Lys in the tail of HDA-1 is reasonable. Likewise, the finding that HDA-1 sumoylation promotes association with NuRD complex components and association of MEP-1, an HDA-1 interactor, with chromatin regulators is convincing. In addition, the evidence that HDA-1 sumoylation increases H3K9ac deacetylation in vivo, leading to negative regulation of hundreds of target genes, and plays a role in the inherited RNAi pathway is solid.While the overall conclusion provides an interesting advance in understanding mechanisms of piRNA-mediated gene silencing in *C. elegans*, the paper is lacking any biochemical analysis of the effects of sumoylation on HDA-1 activity and its association with other transcriptional regulators.1. The authors mapped two sumoylation sites close to the C terminus of HDA-1, K444 and K459, based on extremely weak homology with two established sumoylation sites in human HDAC1 that are reported to be important for transcriptional repression (N.B. the authors should indicate here that David et al. reported that K444/476R HDAC1 had reduced transcriptional repression activity in reporter assays.).

**We thank the reviewer for raising this point. We have added to the discussion on this interesting vertebrate work.**

While the two human sites conform to the sumoylation site consensus, ψKXE, neither K444 nor K459 in HDA-1 fits this consensus (possibly one could argue that K444 is in an inverted motif). The fact that the KKRR mutant HDA-1 is no longer sumoylated is consistent with these two Lys being sumoylated, but it would be reassuring to have direct MS evidence that K444 and K459 are indeed sumoylated, which could be achieved using a SUMO Thr91Arg mutant that generates a GlyGly stub upon trypsin digestion, among other methods.

**As noted above and in the PIE-1 paper, we attempted these experiments. However, mass spec detection of SUMO-modified peptides requires a smo-1 L88K mutant, which is inviable.**

2. It remains unclear how sumoylated HDA-1 is recognized by MEP-1 for assembly into the NuRD complex. Does MEP-1, or another NuRD subunit, have a SIM that could facilitate direct interaction of MEP-1 and sumoylated HDA-1?

**This point is now discussed and was addressed in the earlier comments above.**

3. As the authors discuss, it is surprising that the HDA-1(KKRR)::SUMO protein, which in effect is a constitutively sumoylated form of HDA-1 that will interact constitutively with MEP-1/NuRD, does not have more deleterious effects on the organism, since according to the data in Figure 2B, the stoichiometry of endogenous HDA-1 sumoylation was extremely low. Of course, low sumoylation stoichiometry, which is a general issue with sumoylation studies, means that only a very small fraction of the HDA-1 endogenous population will be able to engage with the silencing complexes at any one time. This point is also worth discussion.

**This too was addressed above.**

4. Page 5: Here, and elsewhere, the authors claim that sumoylation of the two C-terminal Lys activates HDA-1 histone deacetylase activity, but provide no direct evidence for this statement. There are no HDAC assays, and it is unclear how C-terminal SUMO residues distant from the catalytic domain would alter its enzymatic activity, unless there is a SIM motif in HDA-1 that might allow for intramolecular interaction with SUMO residues at the tail leading to a conformation change. Did the authors check for a SIM motif in HDA-1? The fact that adding SUMO to the C-terminus rather than one or both of the two Lys would also have to be taken into account in determining bow sumoylation might "activate" HDA-1. To demonstrate that sumoylation activated HAD-1 in vitro deacetylation assays would need to be carried out comparing the activities of unmodified and sumoylated HDA-1. Instead of enzymatic activation, it is possible that the PIE-1 interaction and HDA-1 sumoylation results in relocalization of HDA-1 within the nucleus to facilitate more efficient H3K9ac deacetylation.

**We apologize again for our careless use of the word “activates.” As noted above, we have changed the manuscript to correct this. It is very unlikely that the enzyme itself is more active but rather that its access to substrate is increased due to interactions with the NuRD components and the Argonaute protein WAGO-9.**

Reviewer #2:In their manuscript, Kim et al. describe a role for HDAC1 (HDA-1) sumoylation in Argonaute-directed transcriptional silencing. The authors suggest that sumoylation of HDA-1 is important for proper assembly of the NuRD deacetylase complex. The role of SUMO modification in heterochromatin has been extensively documented and it is a very interesting topic. The current manuscript provides a very interesting set of results on this topic. I have list of comments, suggestions, questions, and concerns, which are listed below, especially related to the first half of the results:1. A general question would be how can HDA-1 sumoylation, which is barely detectable, account for such a big 'positive' effect on complex assembly? HDA-1 SUMO modification seems around 10% after enriching for SUMO-modified proteins, which means that stoichiometry will be way lower than this. While this is common for SUMO-modified proteins, it does make it difficult to associate with a 'simple' model.

**This issue is now a major discussion point. Thank you! We were so thrilled to even detect the modification in vivo – which is a challenge, as you know – that we neglected to comment on its elusive nature in biochemical studies. We hope the new discussion sufficiently emphasizes this point.**

2. In Figure 1, a schematic of the sensor used throughout the study would benefit the reader.

**We now include a schematic in Figure 1A.**

3. In Figure 1, have the authors checked if the 10xHis::tagged smo-1 has the same effect as the 3xflag::smo-1 (i.e. is it also a partial loss of function allele)?

**Thank you for raising this point. This strain is fully wild type. We now make this point in the PIE-1 paper.**

4. In Figure 1 it would be nice to see the global SUMO conjugation levels in the different conditions, particularly in the smo-1(RNAi), 3xflag::smo-1, and ubc-9(G56R).

**In Figure 3A of the PIE-1 paper, we now show that *smo-1(RNAi)* broadly reduces global SUMO levels.**

5. Also Figure 1, was gei-17 depletion/deletion checked in any way (i.e. WB)? Did the authors consider other SUMO E3 ligase, such as the mms-21 orthologue?

**Thank you for raising this point. The *gei-17* genetics are addressed in the PIE-1 paper. The degron allele does not behave like a null, perhaps because TIR1 is only expressed in the germline (and so GEI-17 still functions in the soma), or because GEI-17 is not completely depleted in the germline by the auxin-inducible degron system. We could not directly measure the level of depletion of GEI-17 in the germline due to the remaining somatic GEI-17. So we removed it from the paper.**

**A null allele of *gei-17* failed to desilence the piRNA sensor. By contrast, the double between the *gei-17* null and *pie-1(K68R)* desilenced the sensor in 100% of isolates.**

**We have not explored other E3s. Given the very strong synthetic phenotype between K68R and *gei-17*, however, we believe that GEI-17 is the most relevant E3. GEI-17 was also identified as a PIE-1 interactor in our two-hybrid screen.**

6. While I am not a big fan of fusing SUMO to proteins, in this case it seems like a very reasonable thing to do, considering the modification sites are located very close to the C-terminal end of the protein. Did the authors check an N-terminal fusion?

**We have not checked the N-terminal fusion. We were surprised this worked too, and only later after doing this did we notice the paper showing that expressing HDA-1::SUMO in the mouse model for Alzheimers improved cognitive function. It seems like HDA-1 may be ideal for this approach since the lysines are so near the C-terminus.**

7. In Figure 2B, it becomes very clear that the level of SUMO modification of HDA-1 is extremely small, barely detectable after an enrichment method. I also wonder why the gels were cropped so tightly, especially considering that in Figure 3 there is an additional band corresponding to ubiquitylated, sumoylated HDA-1. in vitro modification assays would be helpful. HDA-1 alongside a known and characterised SUMO substrate would indicate how good a substrate HDA-1 is.

**We were only trying to conserve space. We have expanded the cropped region of the blot to show the higher molecular weight region. Ubiquitin-SUMO-HDA-1 is not detected. We worry that in vitro approaches would not be enlightening. There are so many other factors that would likely be absent from our in vitro assays. We feel that our genetic and molecular studies provide clear evidence for HDA-1 SUMOylation and its importance in vivo. Indeed, we think that the low level of modification detected make our story more interesting, not less. We suspect that the labile nature of the SUMO conjugation creates a “detection” issue that could explain the apparent absence of HDA-1 from a MEP1-Mi2 complex (called MEC) in a fly ovarian cell line. We now discuss this issue in both papers.**

8. In Figure 2D, is the difference between HDA-1(KKRR)::SUMO and HDA-1::SUMO significant?

**Thank you for raising this point. No, there is no significant difference between them.**

**WE now show this in Figure 2D.**

9. In Figure 3A-C, it would be useful to control whether the GFP::HDA-1 fusion behaves as the untagged one in the sensor assay (wt vs. KKRR).

**Unfortunately, our sensor is a GFP reporter, so we would need to rebuild and validate a different sensor to do this. However, the GFP tag is inserted into the endogenous *hda-1* gene, which is essential. The tagged worms are viable and healthy, suggesting it is a fully functional fusion protein.**

10. I have a few questions regarding Figure 3D:i. Considering the extremely low level of HDA-1 sumoylation, did the authors detect SUMO and ub conjugated HAD-1 (not the SUMO usion)?

**We detected SUMOylated HDA-1 under denaturing conditions. In IP buffer (non-denaturing), SUMO was quickly removed from proteins. We now show this in the PIE-1 paper. We did not detect Ub-modified HDA-1 in HDA-1 IPs. Like SUMO, Ubiquitin conjugation might be labile in IP buffer or the addition of Ub to the SUMO fusion could be an artifact.**

ii. Is ub conjugated to SUMO or to HDA-1?

**We don’t know. We discuss the possibility that SUMO is ubiquitinated because the SUMO antibody does not detect the ubiquitinated HDA-1::SMO-1, suggesting that the SUMO epitope is masked by Ub.**

iii. Does MEP-1 contain any obvious SIMs and or UIMs?

**Yes, MEP-1 has two consensus SIM motifs, as discussed above. We now discuss this in both papers.**

iv. To make a stronger case for the SUMO-dependent interaction model, in vitro interaction assays with recombinant proteins would be extremely useful.

**We are indeed interested in understanding the nature of the physical interactions—especially why SUMO is needed only in the adult. However, such a discovery would merit its own paper and would likely require months if not years of additional work. In short, we do not think SUMO alone is responsible! Some unknown factor in the adult germline may be interfering with MEP-1:HDA-1 binding. Until this factor is identified, in vitro work is unlikely to be informative. In fact, in embryos, HDA-1 is not SUMOylated AND it interacts robustly with MEP-1. So, we expect HDA-1 to interact with MEP-1 in vitro, with or without SUMO.**

11. In the discussion, the authors compare the lack of requirement for GEI-17 in their manuscript with the requirement for Su(var)2-10 in flies. It is very important to back this claim that the authors control GEI-17 depletion (as pointed out in 5).

**Thanks. We addressed this point above.**

Reviewer #3:This manuscript by Kim et al. describes a role of SUMOylation in Argonaute-directed transcriptional silencing in *C. elegans*. The Authors found that SUMOylation of the histone deacetylase HDA-1 promotes its interaction with both the Argonatue target recognition complex as well as the chromatin remodeling NuRD complex. This enables initiation of target silencing. Impaired SUMOylation of HDA-1 leads to loss of interactions with several protein complexes, reduced silencing of piRNA targets, and reduced brood size. While the findings and claims are interesting, some of the novelty is overemphasized and some of the claims are not fully supported by the data.

We have done our best to acknowledge other work on SUMO and its role in piRNA silencing. We apologize if it seems we were over-emphasizing our work at the expense of others. We have expanded the Discussion to improve the context of our work. The comment regarding our claims and data will be addressed below.

1. The importance of HDA-1 SUMOylation for transcriptional repression. The title "HDAC1 SUMOylation promotes Argonaute directed transcriptional silencing in *C. elegans*" implies a central role of SUMOylation in piRNA-mediated transcriptional silencing. The Argonaute HRDE-1/WAGO-9 targets countless transposons as shown previously and also in this manuscript (Figure S3), and so do the HDA-1 degron and Ubc9 mutant, indicating that histone deacetylation and protein SUMOylation are essential processes in TE silencing. However, the HDA-1 SUMOylation mutant (KKRR) only slightly affects 6 TE families (Figure S3), indicating that SUMOylation of HDA-1 might not be a key mediator of this process.

We have revised the manuscript to make this point more clearly. Gene silencing by the piRNA pathway includes two stages, initation and maintenance. The Piwi Argonaute PRG-1 is required for initiation not maintenance. While this role for PRG-1 in intiation was pieced together from transgene studies, the idea is that for most transposons, PRG-1 is/was required only to initially silence the transposon. Afterward the WAGO system maintains silencing indefinitely over many generations. We now explain this important detail better in the Results section.

Furthermore, the authors write that "Our findings suggest how SUMOylation of HDAC1 promotes the recruitment and assembly of an Argonaute-guided chromatin remodeling complex to orchestrate de novo gene silencing in the *C. elegans* germline.", but then they also state that "Comparison with mRNA sequencing data from auxin-treated degron::hda-1 animals revealed an even more extensive overlap with Piwi pathway mutants (Figure S2B), indicating that HDA-1 also promotes target silencing independently of HDA-1 SUMOylation."

We apologize for this confusing error. We sometimes think of the WAGO pathway as being a downstream part of the Piwi pathway, but to do so is confusing in this context. We revised this section to clarify the issue as follows:

“The silencing defect was more severe in auxin-treated degron::hda-1 than in hda-1[KKRR] (**Figure 5—figure supplement 1B and Figure 5—figure supplement 2**), resulting in the increased expression of many more transposons and a more extensive overlap with genes upregulated in rde-3 mutant worms (**Figure 5—figure supplement 1C and Figure 5—figure supplement 2**). This result indicates that HDA-1 also promotes the maintenance of silencing independently of HDA-1 SUMOylation.”Based on their results and their own interpretations, I find that the importance of HDA-1 SUMOylation in piRNA-dependent transcriptional silencing is overemphasized.

Thanks for your very candid comments. I hope the above corrections help you understand the assay used here. The similarities between *prg-1* and *hda-1(KKRR)* from our mRNA-seq data are very striking.

Additionally, the model (Figure 7) implies that for initiation of silencing WAGO recruits HDA-1 to targets. This should be tested by analyzing HDA-1 distribution over WAGO targets in WT and upon loss of WAGO.

As we hope is now clear, our model is that HDA-1 is transiently SUMOylated during the initial phase of converting an active heavily acetylated target gene into a silent state. The distribution of HDA-1, most of which would not be SUMOylated in any case, would be unlikely to provide any new insight.

2. The mechanistic role of HDA-1 SUMOylation.On page 17 (amongst other places) the authors claim that "The SUMOylation of HDA-1 promotes its activity, while also promoting physical interactions with other components of a germline nucleosome-remodeling histone deacetylase (NuRD) complex, as well as the nuclear Argonaute HRDE-1/WAGO-9 and the heterochromatin protein HPL-2 (HP1)".• Regarding activity: Loss of deacetylation/silencing in the SUMO mutant might be due to loss of enzymatic activity, but it might also be due to defects in recruitment/complex formation. There is no data that proves altered enzymatic activity.

We apologize again for the poor word choice. As stated above, we completely agree, and we did not mean to imply that SUMOylation regulates HDA-1 enzymatic activity. We have corrected this mistake throughout both papers.

In fact, Figure 6 indicates SUMO-dependent interaction of WAGO-9 with HDA-1, implying that recruitment is affected. To distinguish between activity and recruitment, at the very least, the authors would need to show that HDA-1 localization to its genomic targets is unaltered upon mutating its SUMOylation site (ChIP-seq of wt and KKRR mutant), while H3K9ac is increased (K9ac ChIP-seq in wt and KKRR mutant) in the mutant. This, in combination with HDA-1 localization in wt and WAGO-9 loss would imply whether complex formation to recruit HDA-1 or HDA-1 enzymatic activity is mostly affected by SUMOylation.

We hope it is now completely clear that we are not proposing that the SUMO modification alters enzyme activity. As detailed above, we have extensively revised the Discussion to address the above issues.

• Regarding physical interactions: Figure 3D shows that if we fuse a SUMO residue to HDA-1, it will interact with MEP-1, while SUMOylation deficient HDA-1 mutant doesn't interact. However, for the WT HDA-1 control, we only see unSUMOylated protein interacting with MEP-1. Furthermore, in the MEP-1 IPs of samples that should contain SUMO-fused HDA-1, the authors detect a lot of "cleaved", unSUMOylated HDA-1. Unless cleavage happened after IP, during elution (unlikely, and there is "cleaved" HDA-1 in the inputs), these findings argue that the interaction with MEP-1 is not mediated by HDA-1 SUMOylation. An interaction between MEP-1 and unmodified HDA-1 is also shown in the accompanying manuscript, which appears to be dependent on Pie-1 SUMOylation. Thus, SUMOylation of HDA-1 alone seems unlikely to be the major factor necessary for silencing complex assembly. (as a side question: Does the protease inhibitor cocktail used inhibit de-SUMOylation enzymes? I am concerned that deSUMOylating enzymes might compromise some result interpretations.)

Thanks again for your very critical read of our findings. As noted in the cover letter, we have dealt with the fleeting nature of this modification for so many years that—while developing this story—we forgot that it was not generally known or appreciated. In both studies, the SUMO modifications are completely removed when we prepare samples in IP buffer (even 5 minutes at 4°C). We only detect SUMO-conjugated proteins under stringent denaturing conditions. We now make a major discussion point of this interesting and perhaps underappreciated aspect of SUMO biology.

• Regarding functional relevance of HDA-1 acetylation: On pages 12/13 authors claim that because "HDA-1(KKRR) animals and mep-1-depleted worms revealed dramatically higher levels of H3K9Ac compared to wild-type" and "HDA-1, LET-418/Mi-2, and MEP-1 bind heterochromatic", "SUMOylation of HDA-1 appears to drive formation or maintenance of germline heterochromatin regions of the genome." These correlations do not prove function. The authors have performed H3K9me2 (although not H3K9-ac) ChIP-seq in WT, KKRR mutant and HDA-1 degron worms, yet do not analyze globally whether acetylation is lost on genes that are affected (change in RNA-seq vs. change in K9me2 or acetyl). To support the claim that SUMOylation of HDA-1 drives deacetylation and heterochromatin formation, it would be important to show changes in H3K9Ac levels (or other acetyl marks) and potentially NuRD component occupancy between control and HDA-1 SUMOylation-deficient animals at specific targets (i.e. genes derepressed upon loss of SUMOylation identified in RNA-seq, and the reporter locus).

As we note above, obtaining sufficient material for ChIP-seq studies on adult germline is technically difficult to do.

3. The authors claim (p17) that "initiation of transcriptional silencing requires SUMOylation of conserved C-terminal lysine residues in the type-1 histone deacetylase HDA-1". I do not see any supporting data that has separately looked at formation/initiation and maintenance of silencing (a technically challenging experiment).

Thanks again for pointing this out. The piRNA pathway initiates transcriptional silencing on actively transcribing genes, and it is in this context where HDA-1 SUMOylation appears to be most critical. We hope this issue is now explained better in the manuscript.

4. The authors repeatedly claim that gei-17 does not play a role in piRNA target silencing, based on loss of gei-17 not affecting the piRNA reporter (Figure 1B). At the same time, they claim that pie-1 plays a role, even though it likewise does not affect the piRNA reporter (it affects the reporter only in F3; data on gei-17 effect in F3 is not present). In the accompanying paper, the authors show that while gei-17 loss by itself causes only moderate effect on extra intestine cells, combined with Pie-1 loss the effect is more severe than when Pie-1 loss is combined with Ubc9 or smo loss. This to me indicates an important role of gei-17 in inhibiting differentiation of germline stem cells to somatic tissues, but these effects are likely synergistic and thus masked by Pie-1. Individually neither Gei-17 nor Pie-1 show an effect on piRNA reporter in P0, but to confirm lack of synergy, their effects of should be tested together. Although possible, the present data is insufficient to rule of gei-17 involvement.

You are correct. We have now completed double mutant studies that show that *gei-17* and *pie-1* function together to silence the piRNA sensor. This new data is in presented the PIE-1 paper, and discussed in both papers.